# Segmental motor recovery after cervical spinal cord injury relates to density and integrity of corticospinal tract projections

Gustavo Balbinot [1,2,3] ✉, Guijin Li [1,4], Sukhvinder Kalsi-Ryan [1,5,6], Rainer Abel [7], Doris Maier[8], Yorck-Bernhard Kalke [9], Norbert Weidner[10], Rüdiger Rupp [10], Martin Schubert [11], Armin Curt[11] & Jose Zariffa [1,4,5,12] ✉

Cervical spinal cord injury (SCI) causes extensive impairments for individuals which may include dextrous hand function. Although prior work has focused on the recovery at the person-level, the factors determining the recovery of individual muscles are poorly understood. Here, we investigate the muscle-specific recovery after cervical spinal cord injury in a retrospective analysis of 748 individuals from the European Multicenter Study about Spinal Cord Injury (NCT01571531). We show associations between corticospinal tract (CST) sparing and upper extremity recovery in SCI, which improves the prediction of hand muscle strength recovery. Our findings suggest that assessment strategies for muscle-specific motor recovery in acute spinal cord injury are improved by accounting for CST sparing, and complement person-level predictions.

Spinal cord injury (SCI) is defined as damage to the spinal cord resulting in temporary or permanent changes in its function[1]. SCI poses important physical and social consequences for the affected individuals, and the care of individuals with SCI requires substantial efforts. The development of effective treatments becomes crucially important to enhance spinal cord function, and subsequently, improve sensorimotor function and minimize secondary complications. Understanding sensorimotor recovery under the current standard of care enables the identification and prediction of persistent functional impairments that may be improved by treatment, and is essential to accurately assess the impact of interventions. In tetraplegia, the improvement in upper limb motor function is important and regaining hand function is considered a high priority[2]. Although extensive effort has been devoted to understanding recovery of upper extremity function and strength[3–12], little is known about how the segmental innervation of upper limb muscles

recovers after SCI[13]. It is known that the impairment of upper limb muscles is related to task performance[8,14] and assessing strength of upper extremity muscles enables to predict upper limb function[15]. Nonetheless, the specific recovery profile of single upper limb muscles is still poorly understood, especially that of the hand muscles[13].

Several factors may contribute to variations in recovery profiles across upper limb muscles. Upper limb muscles are controlled by integrated and relatively overlapped representations in the motor cortex[16], but the cortical representation of hand muscles is larger, with extensive corticospinal tract (CST) connections to cervical spinal motoneurons[17–19]. It is thought that spinal motoneurons of the distal compared to proximal upper limb muscles receive greater input from the primary motor cortex through the CST to execute more refined, versatile fine movements[20–22]. When the spinal cord is injured, motor tracts may be damaged affecting the integrity of the CST[23–27]. The role

[1]KITE, Toronto Rehabilitation Institute, University Health Network, Toronto, ON, Canada. [2]Krembil Research Institute, University Health Network, Toronto, ON, Canada. [3]Center for Advancing Neurotechnological Innovation to Application – CRANIA, University Health Network, Toronto, ON, Canada. [4]Institute of Biomedical Engineering, University of Toronto, Toronto, ON, Canada. [5]Rehabilitation Sciences Institute, University of Toronto, Toronto, ON, Canada. [6]Department of Physical Therapy, University of Toronto, Toronto, ON, Canada. [7]Hohe Warte Bayreuth, Bayreuth, Germany. [8]BG-Trauma Center, Murnau, Germany. [9]RKU Universitäts- und Rehabilitationskliniken Ulm, Ulm, Germany. [10]Spinal Cord Injury Center, Heidelberg University Hospital, Heidelberg, Germany. [11]Spinal Cord Injury Center, Balgrist University Hospital, Zurich, Switzerland. [12]Edward S. Rogers Sr. Department of Electrical and Computer Engineering, University of Toronto, Toronto, ON, Canada. ✉e-mail: gustavo.balbinot@hotmail.com; jose.zariffa@utoronto.ca

of CST integrity on upper limb motor recovery is a topic extensively studied in stroke. The relationship between motor recovery and the initial impairment reflects the biological limits of structural and functional plasticity. Individuals with a stroke severely affecting CST integrity display great impairment and limited recovery of upper limb function and do not fit the proportional relationship that has been observed between the amount of recovery and the initial impairment in individuals with less CST damage[28,29]. It is known that 'non-fitters' have limited performance in tasks related to wrist/hand dexterity, which is also indicative of a more pronounced CST disruption[30]. In SCI, the lesion will often affect the projections from the CST and other descending tracts to spinal motoneurons (i.e., axonal lesions of upper motor neurons; UMN) and/or directly damage α-motoneurons [i.e., lower motor neuron (LMN) lesion], depending on the extent, location, and severity of the lesion. UMN versus LMN damage is not distinguished by clinical exam (e.g., International Standards for Neurological Classification of Spinal Cord Injury; ISNCSCI)[31]. Thereby, the effect of UMN and LMN lesions may vary across upper limb muscles; for anatomical reasons CST damage may have more pronounced effects on distal movements, while LMN damage is likely to be more pronounced at or immediately below the lesion. In addition, the upper limb is comprised of muscles specialized for both gross and fine motor function, leading to variations in the number and size of motor units and muscle fiber types across muscles[32,33]. The impact of such variations on muscle functional recovery in SCI is poorly understood.

Established recovery profiles after SCI have not distinguished the development of spastic or flaccid muscles weakness, and summed motor scores (i.e., upper extremity motor score; UEMS) do not discern the recovery of distal or proximal upper limb muscles. Although it is known that the residual muscle strength early after SCI is indicative of preserved CST connections and a good predictor of summed upper limb strength recovery[6,7], the prediction of individual myotomes is lacking. Neurophysiological assessments such as motor evoked potentials (MEPs), somatosensory evoked potentials (SSEPs) and compound muscle action potential (CMAPs) have been applied to assess CST and/or α-motoneuron integrity and the natural extent of spinal neural recovery contributing to the prediction of gross functions like walking and independence[4], and may be beneficial as well for predicting myotome recovery. Here, the muscle-specific approach supports an emerging scenario in the field aimed at better understanding motor discomplete lesions[34–37], and the importance of lateral tract sparing[38,39] and the zone of partial preservation (ZPP)[40] for recovery prognostics.

In this work, we explore if segmental innervation as assessed in single upper limb muscles exhibits different strength recovery profiles after cervical SCI. We aim to identify factors predictive of segmental strength recovery in upper extremity muscles and hypothesize that neuroanatomical factors as related to segmental muscles (i.e., extent of corticospinal connections, and distance to the motor level of lesion) may affect the potential for recovery. We show that the recovery after cervical SCI follows a proximal-to-distal gradient in which distal muscles of the upper limb show limited and delayed strength recovery compared to proximal muscles. The motor recovery of the hand muscles is also hard to predict and the addition of baseline features related to CST integrity enhances such predictions. Overall, our data support the importance of CST and LMN integrity in indicating spinal cord dysfunction and recovery, here evidenced by the residual strength and MEP at baseline. Our findings partially mirror observations of people recovering from stroke indicating that post-stroke individuals with MEP+ recover ≈70% of lost upper extremity function[28,29]. Here, we show that post-SCI individuals with MEP+ recover ≈22–45% of what they have lost – a proportion much lower compared to stroke, without any clear separation into "fitters" and "non-fitters" commonly seen in stroke.

## Results

The clinical records of 748 research participants were reviewed in this study (26/748 with non-traumatic SCI) (Fig. 1). There were 599 males and 149 females, 261 classified as AIS A, 84 as AIS B, 155 as AIS C, and 241 as AIS D[31]. Neurological level of injury ranged from C1 to C8 and the mean age was 46.5 years. Very acute clinical assessments were available for 440 participants and multimodal electrophysiological assessments were conducted in 203 (MEPs), 313 (SSEPs), and 280 (NCS) participants at the 4-week baseline (Supplementary Table 1).

In a first step, we describe the segmental strength recovery profiles after cervical SCI. In a second step, we investigate the ability of several baseline anatomical and injury characteristics (summarized in Fig. 2) to predict segmental recovery.

### Segmental strength recovery in upper limb muscles

In accordance with previous findings[41], individuals with AIS A and AIS B lesions have similar median UEMS early after the lesion (1w–4w) but participants with an AIS B regain more UEMS with time, 7 points at 12w ($p = 0.003$, U = 2823), 9 points at 24w ($p = 0.004$, U = 2865) and 12 points at 48w ($p = 0.001$, U = 2726)). Individuals with an AIS B and C display similar upper limb recovery profiles ($p > 0.008$), but AIS C show greater UEMS at all time points, compared to AIS A (1w: 7 points, $p = 0.002$, U = 5788; 4w: 9 points, $p < 0.001$, U = 5013; 12w: 12.5 points, $p = 0.001$, U = 4178; 24w: 18 points, $p < 0.001$, U = 3962; 48w: 19 points, $p < 0.001$, U = 2847). Participants with an AIS D display the most upper limb strength recovery ($p < 0.05$; Fig. 3a, b provide patterns of absolute UEMS scores and statistical comparison results, respectively).

At the segmental muscle level, after controlling for the distance from the motor level (DST; Fig. 3c), proximal muscles such as the elbow flexors and extensors show superior recovery compared to distal muscles such as the intrinsic hand muscles, especially if distant from the lesion. For example, at a DST of −3, the elbow flexors recover to a grade of 3 or more in 77.8% of individuals classified as AIS A and 100% of individuals classified as AIS D (Fig. 3d); the wrist extensors recover to a grade of 3 or more in 51.1% of individuals classified as AIS A and 100% of individuals classified as AIS D (Fig. 3e); the elbow extensors recover to a grade of 3 or more in 21.4% of individuals classified as

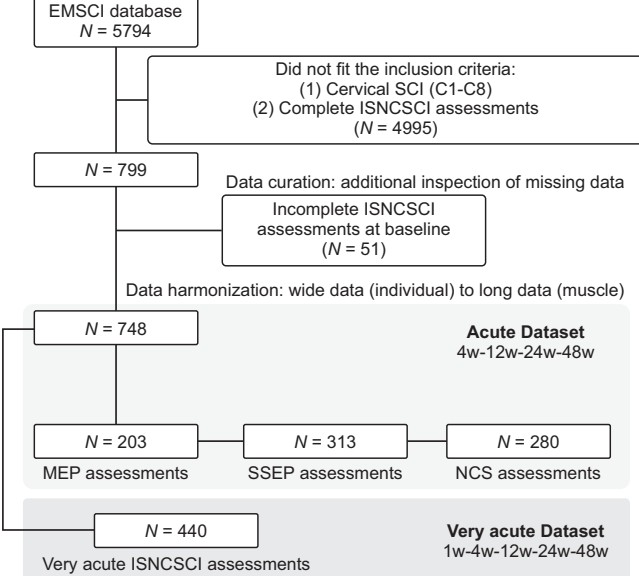

**Fig. 1 | Participants included from the EMSCI Dataset.** EMSCI = European Multi-Center Study about Spinal Cord Injury, ISNCSCI = International Standards for Neurological Classification of Spinal Cord Injury, SCI = Spinal Cord Injury, w = Week, MEP = Motor Evoked Potential, SSEP = Somatosensory Evoked Potential, NCS = Nerve Conduction Studies.

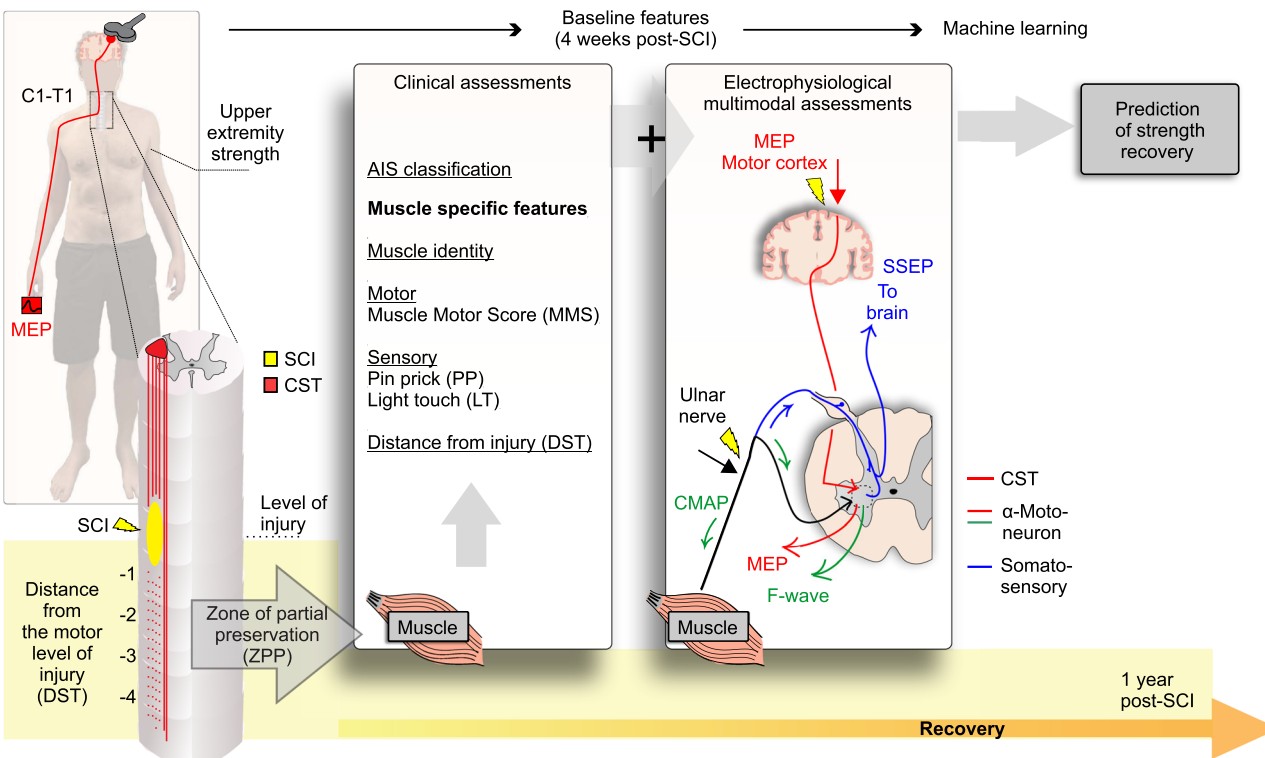

**Fig. 2 | Anatomical and injury characteristics hypothesized to be determinants of segmental muscle recovery.** Muscle identity may play a role due to variations in cortical representation and CST projections. Remaining spinal innervation after injury may be reflected in the muscle motor score (MMS), sensory scores in corresponding dermatomes (PP and LT), and electrophysiological assessments (MEP, SSEP). NCS may improve predictions by providing information about α-motoneuron damage. AIS grade and distance from injury further determine the capacity for neurorecovery. C = Cervical, T = Thoracic, SCI = Spinal Cord Injury, CST = Corticospinal Tract, AIS = American Spinal Cord Injury Association Impairment Scale, MEP = Motor Evoked Potential, SSEP = Somatosensory Evoked Potential, NCS = Nerve Conduction Studies, MMS = Muscle Motor Score, ZPP = Zone of Partial Preservation, DST = Distance from the motor level of injury, LT = Light Touch sensation, PP = Pin Prick sensation.

AIS A and 95.4% of individuals classified as AIS D (Fig. 3f); the finger flexors recover to a grade of 3 or more in 18.5% of individuals classified as AIS A and 96% of individuals classified as AIS D (Fig. 3g); the finger abductors recover to a grade of 3 or more in 29.7% of individuals classified as AIS A and 85.7% of individuals classified as AIS D (Fig. 3h). Overall, muscles from individuals classified as AIS A also take longer to recover strength to a grade 3 or more ($p < 0.05$). Although some statistical comparisons are hampered due to the low sample size of the very acute dataset (especially for AIS B/C – Supplementary Fig. 1), additional analysis using the complete dataset (not including the 1w timepoint) corroborate these findings (Supplementary Figs. 2–5). Finally, we performed a dataset split into muscles from the left and right sides of the body. This analysis indicated that our findings were not influenced by the left and right merging procedure used for the segmental analysis (Supplementary Fig. 6). The bulk of the results indicates greater impact of the SCI and lesser strength recovery of distal muscles (finger flexors and abductors) compared to proximal upper limb muscles (elbow flexors), especially in individuals with sensorimotor or motor complete SCI, even after controlling for the distance from the lesion.

**Strength recovery after cervical SCI: the role of baseline muscle motor score (MMS)**
Considering all AIS grades and all upper limb muscles, the prediction of strength recovery by baseline muscle motor score (MMS) is poor ($R^2 = 0.148$; Fig. 4a). On average, muscle-level analysis indicates that there is some strength recovery (change in MMS scores from 4 to 48 weeks) if the initial impairment is low (1–2), and plateaus between 1 and 2 points for greater initial impairments (Fig. 4a). For individuals

classified as motor complete (AIS A and B), strength recovery on average in the group of analyzed muscles is constant or inverse if the initial impairment is high–especially for hand muscles (Fig. 4b, c). Strength recovery (characterized by a quasi-linear, incremental regression line) is apparent on average in individuals classified as AIS C for elbow flexors, wrist extensors, elbow extensors, and finger flexors. Some recovery is also apparent for finger abductors if the initial impairment is low to mild (1–3) but plateaus at 1.5 points if the initial impairment is high (4–5; Fig. 4d). Baseline MMS predicts strength recovery in AIS D patients, with the strength recovery being evident for all muscles in these individuals (Fig. 4e). Considering the different muscles and AIS grades, the non-linear regression using random forest algorithms using only the baseline MMS indicates good prediction of strength recovery for all muscles of AIS D participants with high $R^2$ values and a prediction error of ≈0.5 points. Although the prediction is fair to good for some of the proximal muscles in individuals classified as AIS A/B/C, predicting strength recovery solely based on the initial motor impairment (baseline MMS) is overall poor, especially for distal muscles ($R^2 ≈ 0.1$) (Fig. 4f). To understand the influence of merging muscles from the left and right sides of the body on the non-linear regression using random forest regressors, we also performed a dataset split into muscles from the left and right sides of the body. This analysis indicated that our findings were not influenced by the left and right merging procedure used for the segmental analysis (Supplementary Fig. 7).

**The role of additional muscle-specific features**
Given the inability to predict strength recovery using solely the baseline MMS in individuals classified as AIS A/B/C, especially in distal

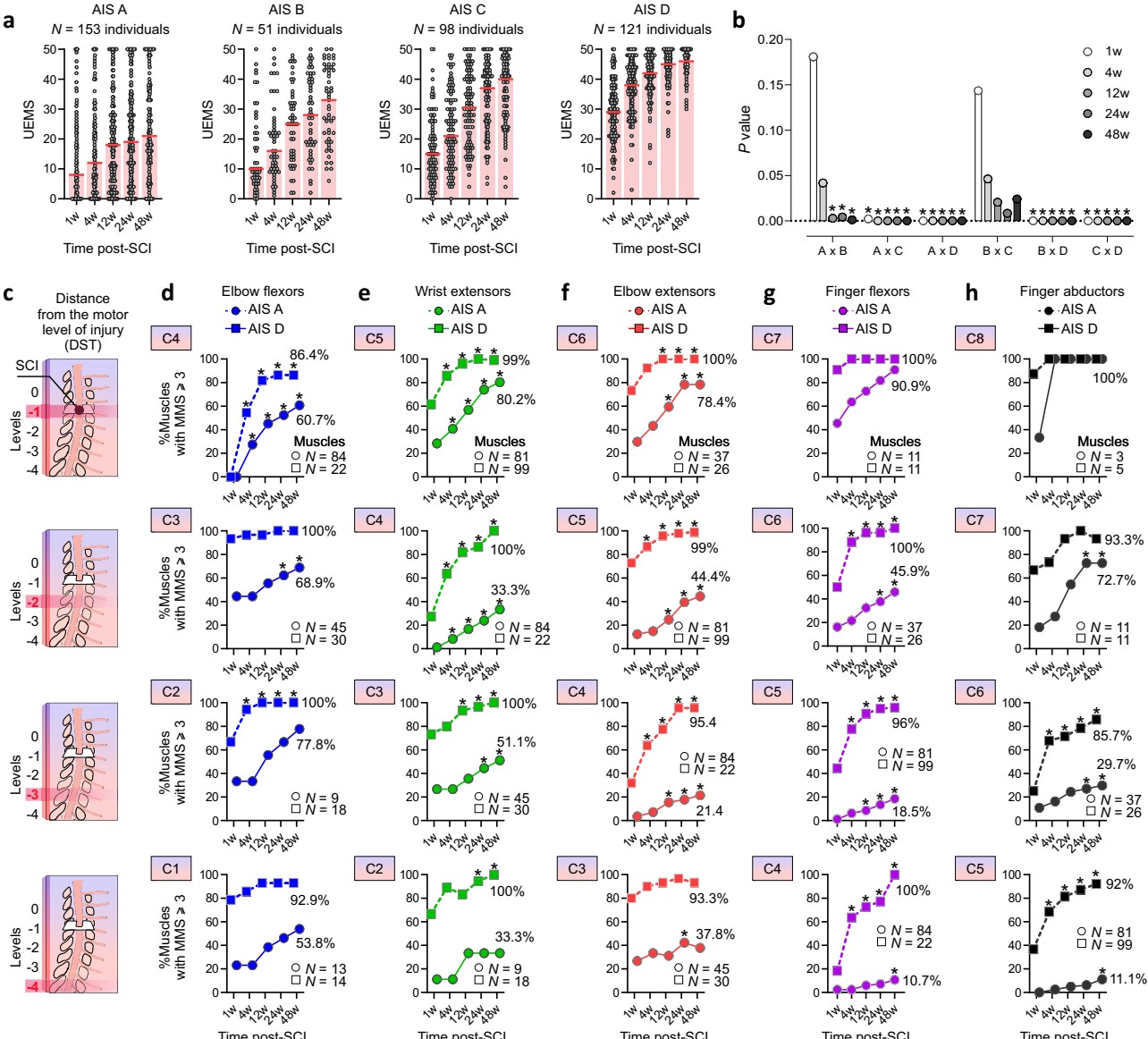

**Fig. 3 | Strength recovery in upper limb muscles after cervical SCI. a** The median UEMS in AIS A-D indicates some recovery independent of the SCI severity. While UEMS at onset are similar in AIS A and AIS B at 1w–4w post-SCI the extent of recovery at 12w–48w is higher in AIS B. AIS C and D show different UEMS at onset and over time. **b** Statistical comparisons for the UEMS data in (**a**). **c** The distance between the motor level and the myotome (DST) was controlled in D–H. **d**–**h** In individuals classified as AIS A, the probability of the proximal muscles (i.e., elbow flexors, wrist extensors, and elbow extensors) achieving against gravity strength (MMS ≥ 3) was greater compared to hand muscles (i.e., finger flexors and abductors)—especially if the hand muscles are distant from the SCI (i.e., levels −3 and −4). Hand muscles also took longer to regain strength in individuals classified as AIS A.

In participants classified as AIS D, the overall probability of upper limb muscles reaching MMS ≥ 3 was greater (≈97%) compared to AIS A (≈ 50.6%), with the lowest probabilities for hand muscles. The center line is the median in (**a**). In (**b**) the plotted *p* values are related to the sample sizes described in (**a**). Data are % of muscles with an MMS ≥ 3 in (**d**–**h**). The insets in (**d**–**h**) describe the motor level of injury (C1–C8). *$p < 0.05$, Multiple Mann–Whitney tests with multiple comparison adjustments using false discovery rate in (**a**, **b**) (two-sided), McNemar's tests (two-sided) in (**d**–**h**). *N* = number of biologically independent samples, SCI = Spinal Cord Injury, AIS = American Spinal Cord Injury Association Impairment Scale, MMS = Muscle Motor Score, UEMS = Upper Extremity Motor Score.

muscles, we explored additional segment-specific variables available in the ISNCSCI [i.e., light touch (LT), pin prick sensation (PP), and the distance from the motor level of injury (DST)]. A four-step approach using supervised machine learning models (Figs. 5a and 6a) is employed to predict the presence or absence of recovery at the muscle level. First, we corroborate the importance of baseline AIS classification and MMS as predictive factors for strength recovery in SCI but expand it to the predictions of segmental strength recovery (Fig. 5b; see Supplementary Table 3 for feature importance in each model).

The classification performance assessed by the precision-recall area under the curve (PR AUC) is overall higher for AIS C/D compared

to AIS A/B, and the addition of the muscle identity as a feature does not afford an increase in the classification accuracy (All AIS: $p = 0.095$; AIS A: $p = 0.944$; AIS B: $p = 0.832$; AIS D: $p = 0.924$) or decreases the classification accuracy (AIS C: $p = 0.003$) (Fig. 5c, d–upper panels). Note that the models for AIS C/D are imbalanced and have limited support for the 'No Recovery' class, leading to a high false positive rate and low performance on the receiver operating characteristic area under the curve (ROC AUC) (Fig. 5c, d–lower panels). When constructing muscle-specific models, the prediction of strength recovery in elbow flexors displays a good performance on the PR AUC; a moderate performance is evident for wrist extensors and elbow extensors, but the prediction

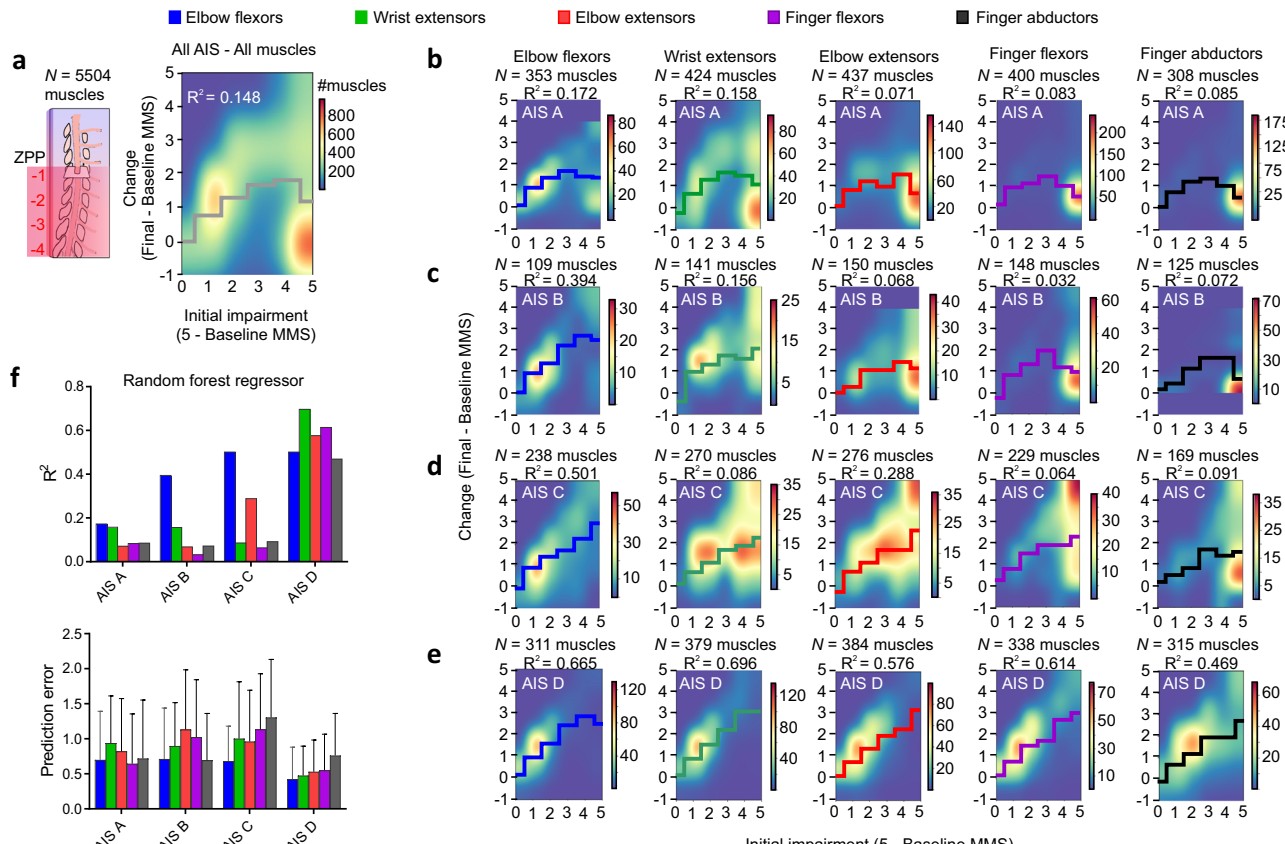

**Fig. 4 | Prediction of strength recovery after cervical SCI: the role of baseline MMS.** Baseline MMS is a good predictor of strength recovery at 1-year post-SCI for individuals with AIS D (high $R^2$ values) but is in most cases a poor predictor for those with an AIS A/B/C. **a** Considering all AIS and muscles, the prediction of strength recovery using baseline MMS is poor ($R^2 = 0.148$). **b, c** Individuals classified as AIS A or B show some degree of proportional recovery of upper limb muscles if the initial impairment is low, but strength recovery is constant or is inversely proportional if the initial impairment is high (especially for distal hand muscles). **d** In individuals classified as AIS C, proportional strength recovery is apparent for elbow flexors, wrist extensors, elbow extensors, and finger flexors. Proportional recovery is also evident for finger abductors if the initial impairment is low to mild (baseline MMS from 3 to 5) but is constant if the initial impairment is high (baseline MMS from 0 to 1). **e** Proportional strength recovery is evident for

all muscles in individuals classified as AIS D. **f** Summary of the non-linear regression using random forest regressors indicates good prediction of strength recovery for all muscles of AIS D participants with a prediction error of ≈0.5 points. Although the prediction is fair to good for some of the proximal muscles in individuals with an AIS A/B/C, predicting late strength recovery solely based on the initial motor impairment is poor for distal hand muscles ($R^2 ≈ 0.1$). Complex analysis using random forest regressor with 50% of the dataset for training and 50% for testing with 100 trees (estimators) in (**a**–**e**). Data are Mean ± SD in (**f**), bottom panel. In (**f**) the plotted $R^2$ and prediction errors are related to the sample sizes described in (**b**–**e**). $N$ = number of biological samples (please note that muscles from the left and right sides of the same individual are pooled in this analysis). SCI = Spinal Cord Injury, AIS = American Spinal Cord Injury Association Impairment Scale, MMS = Muscle Motor Score, ZPP = Zone of Partial Preservation.

of strength recovery is lower for hand muscles (Fig. 5e). Similar to the AIS models described in Fig. 5c, d, the imbalance in support for the 'No Recovery' class of proximal muscles leads to a high false positive rate and low performance on the ROC AUC (Fig. 5e–lower panel). Besides the limitations of the muscle-specific models (imbalance), it is evident that AIS A and B display a proximal to distal gradient, where the strength recovery of hand muscles is hard to predict. In individuals classified as AIS C, the prediction of strength recovery is good for all muscles except the finger abductors. Prediction of strength recovery in participants classified as AIS D shows good performance for all muscles (Fig. 5f). Note that because of the imbalanced datasets, the muscle-specific models for AIS C/D were trained on a reduced number of muscles from the negative 'No recovery' class and perform poorly in predicting it (Fig. 5g, h). Overall, the prediction of strength recovery displays a proximal-to-distal gradient in individuals with a sensor-imotor complete lesion, where the strength recovery of distal muscles is hard to predict despite the inclusion of additional predictive variables in the model (i.e., muscle identity, LT and PP sensation). Although the residual strength at baseline (baseline MMS) is an important feature in predicting strength recovery, especially in distal muscles (data not shown), it is not sufficient to afford a good prediction of strength

recovery for distal muscles in individuals classified as AIS A/B/C (Fig. 5f).

Finally, to understand the effects of merging muscles from the left and right sides, we performed a leave-one-subject-out cross-validation procedure. This additional validation layer ensured that none of the muscles from the test participant were included when training the model, at each fold. This analysis yielded similar results compared to the leave-one-muscle-out cross-validation procedure described in Fig. 5 and is summarized in Supplementary Table 4.

### The role of electrophysiological biomarkers
Next, we sought to understand the predictive value of electro-physiological multimodal assessments in improving outcome prediction of strength recovery of distal muscles in individuals classified as AIS A/B/C (Fig. 6a, b). Overall, biomarkers of CST and LMN integrity and somatosensory integration are increased at baseline in muscles that showed strength recovery 48 weeks after SCI (Table 1). Baseline MEP amplitudes of hand muscles with strength recovery are greater ($p < 0.0001$, $U = 2372$), and those hand muscles tend to have higher MEP scores (which indicates both high amplitude and low latency), compared to muscles with absent recovery ($p < 0.0001$, $U = 2294$). The

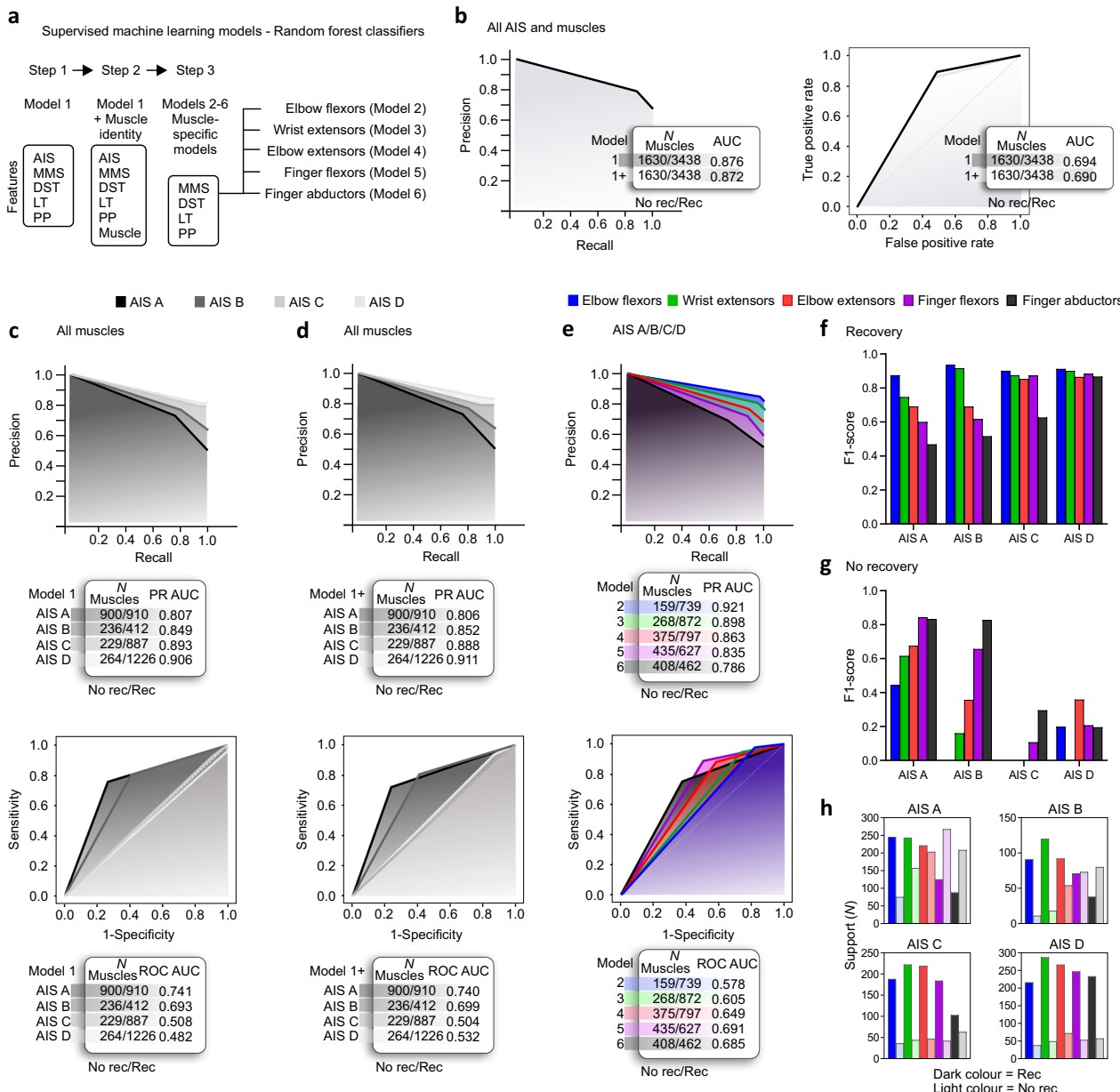

**Fig. 5 | Prediction of strength recovery after cervical SCI: the role of additional muscle-specific features.** The prediction of strength recovery displays a proximal-to-distal gradient in individuals with a sensorimotor complete lesion, where the strength recovery of distal hand muscles is hard to predict. **a** Supervised machine learning models: a three steps approach is utilized to understand the predictive factors for segmental strength recovery after cervical SCI. **b** We corroborate the importance of AIS and MMS in predicting recovery after SCI with a PR AUC of ≈0.87 and ROC AUC ≈ 0.69 (see Supplementary Table 3 for feature importance in each model). **c** AIS-specific models indicate it is harder to predict strength recovery in AIS A/B, compared to AIS C/D. **d** The addition of muscle identity as a feature does not increase the prediction performance of the AIS-specific models. **e** Muscle-specific models indicate a proximal to distal gradient, where the strength recovery of distal hand muscles is harder to predict compared to the proximal muscles. **f** In individuals classified as AIS A, the prediction of strength recovery displays a good performance for elbow flexors, a moderate performance is evident for wrist and

elbow extensors, but the prediction of strength recovery is poor for hand muscles. The prediction of strength recovery is good for elbow flexors and wrist extensors but moderate for elbow extensors and poor for the hand muscles in participants classified as AIS B. In individuals classified as AIS C, the prediction is good for all muscles, except for finger abductors. Prediction of strength recovery in participants classified as AIS D shows good performance for all muscles. **g, h** Note that the muscle-specific models must be interpreted with caution because of the imbalanced datasets. AIS C/D and proximal muscles are trained with a predominance of muscles from the positive class ('Recovery' class), thus, performing poorly in classifying the negative class ('No recovery' class). Complex analysis using random forest classifier with leave-one-muscle-out cross-validation. AIS = American Spinal Cord Injury Association Impairment Scale, AUC = Area Under the Curve, MMS = Muscle Motor Score, DST = Distance from the motor level of injury, LT = Light Touch sensation, PP = Pin Prick sensation, PR = Precision-Recall, ROC = Receiver Operating Characteristic.

SSEP and F-wave persistence of 'Recovery' muscles are also greater, compared to muscles with absent strength recovery 48 weeks after SCI (SSEP amplitude: $p < 0.0001$, $U = 5248$, SSEP score: $p < 0.0001$, $U = 4972$; F-wave persistence: $p = 0.0009$, $U = 4011$). This indicates the spinal cord is more responsive in integrating and transmitting neural

input early after the injury in muscles regaining strength 1 year after SCI. CMAP amplitude was similar between 'Recovery' and 'No recovery' muscle groups, indicating the absence of LMN lesion in spinal segments innervating the *abductor digiti minimi* muscle ($p = 0.316$, $U = 4879$). Although SSEP and MEP assessments improve the

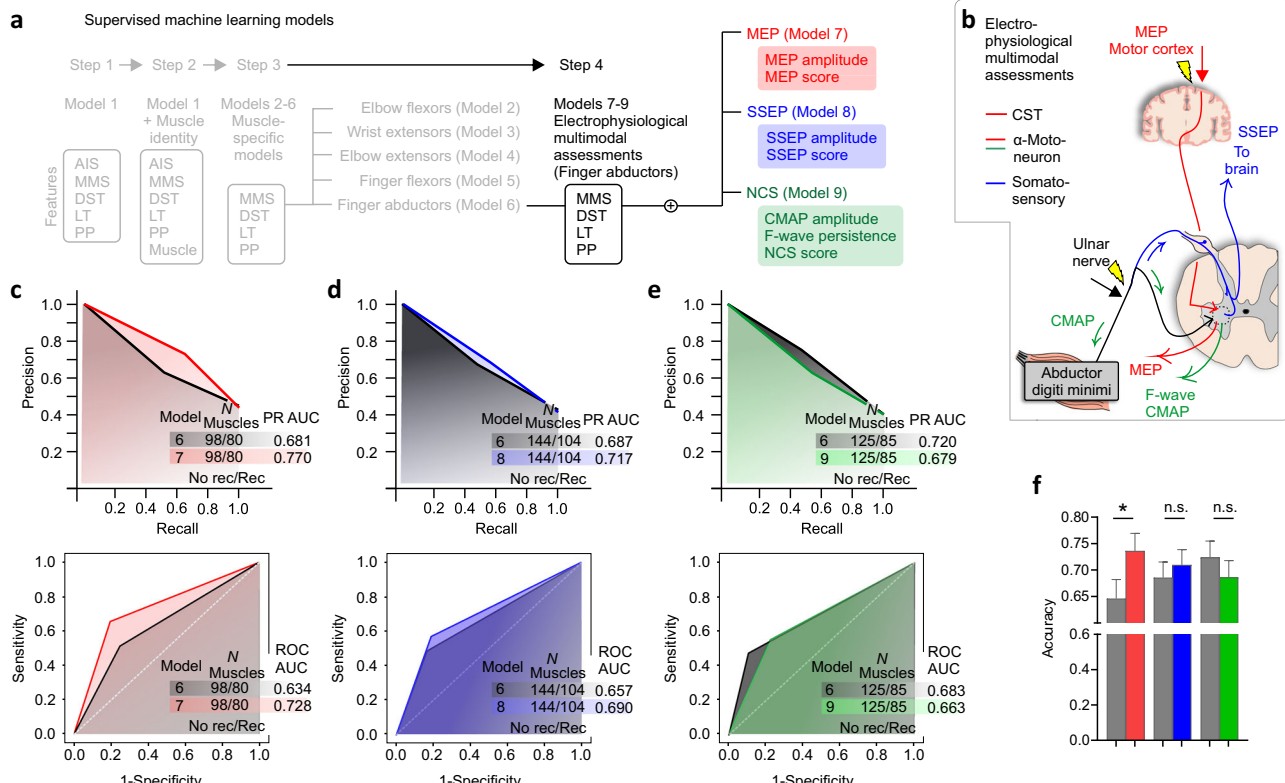

**Fig. 6 | Prediction of strength recovery after cervical SCI: measures of spinal cord function (CST and LMN) integrity increase the classification performance of strength recovery of distal hand muscles in individuals with an AIS A/B/C.** **a** Supervised machine learning models: a fourth step is utilized to understand the predictive factors for strength recovery in finger abductor muscles in AIS A/B/C. **b** Electrophysiological multimodal assessments of MEP, SSEP, and NCS of the distal muscles of the upper limb (finger abductors: *abductor digiti minimi*). MEP amplitude and latency at the *abductor digit minimi* muscle was used to quantify the CST and LMN integrity (red line). SSEP was measured over the scalp after stimulation of the ulnar nerve using needle electrodes (blue lines). Ulnar nerve stimulation was also used during the NCS to measure F-waves and CMAP at the *abductor digit minimi* (green line). **c–e** PR and ROC curves indicate that the overall classification performance assessed by the AUC is increased for the MEP and SSEP subgroups. The most important electrophysiological features are MEP amplitude, SSEP

amplitude, and CMAP amplitude (data not shown). **f** Only the addition of the MEP features afforded a significant increase in the accuracy of the classification (*p* = 0.015). Data are Mean ± SEM in (**f**) to improve visualization. Complex analysis with random forest classifier using leave-one-muscle-out cross-validation in (**c–f**). In (**f**) the plotted accuracies are related to the sample sizes described in (**c–e**). *N* = number of biological samples (please note that muscles from the left and right sides of the same individual are pooled in this analysis). **p* < 0.05, McNemar's test (two-sided) in (**f**). AIS = American Spinal Cord Injury Association Impairment Scale, AUC = Area Under the Curve, MMS = Muscle Motor Score, LT = Light Touch sensation, MEP = Motor Evoked Potential, CST = Corticospinal Tract, DST = Distance from the motor level of injury, SSEP = Somatosensory Evoked Potential, NCS = Nerve Conduction Studies, CMAP = Compound Muscle Action Potential, PR = Precision-Recall, ROC = Receiver Operating Characteristic.

## Table 1 | Outcome of electrophysiological examinations at baseline

| MEP | No recovery | Recovery | U | p |
|---|---|---|---|---|
| Amplitude (mV) | 0 (0–0) | 0.1 (0–0.627)*** | 2372 | <0.0001 |
| Score | 0 (0–0) | 2 (0–3)*** | 2294 | <0.0001 |
| **SSEP** | **No recovery** | **Recovery** | **U** | **p** |
| Amplitude (uV) | 0 (0–1.175) | 1 (0–1.838)*** | 5248 | <0.0001 |
| Score | 0 (0–2) | 2 (0–3)*** | 4972 | <0.0001 |
| **NCS** | **No recovery** | **Recovery** | **U** | **p** |
| CMAP amplitude (mV) | 2.9 (0.6–5.050) | 2.7 (0.835–6.450) | 4879 | 0.316 |
| F-wave persistence (%) | 0 (0–40) | 30 (0–75)** | 4011 | 0.001 |
| Score | 1 (1–2) | 2 (1–3)* | 4487 | 0.0425 |

'Recovery' = a gain ≥1 in MMS at 48 weeks post-SCI, compared to baseline; 'No recovery' = no change or decline in MMS at 48 weeks post-SCI, compared to baseline. Data are median (25–75% percentile).
***p* < 0.001; ***p* < 0.01; **p* < 0.05 Mann–Whitney test (two-sided).

prediction of strength recovery in hand muscles (Fig. 6c–e), only the MEP increased the performance of the random forest classifier significantly (*p* = 0.015; Fig. 6f).

To understand the effects of the dependency in the dataset, which contained muscles from the left and right sides of the same participant, we performed a leave-one-subject-out cross-validation procedure. This analysis indicates the maintenance of the main findings, with a significant increase in the accuracy of the model with the addition of the MEPs (*p* = 0.049; single-tailed), the absence of effect with the addition of SSEPs (*p* = 0.140; single-tailed), and worsening of the prediction with adding data extracted from NCSs (*p* = 0.009; single-tailed) (Supplementary Table 5).

The strength recovery of hand muscles is limited in individuals classified as AIS A/B/C with an MEP⁻ at baseline, while the absence of MEP at baseline was indicative of greater impairment (MMS = 0) and reduced strength recovery in the finger abductors (Fig. 7a). Finger abductor muscles with an MEP⁺ at baseline display greater variability in the initial motor impairment and strength recovery 48 weeks post-SCI, compared to MEP⁻ muscles (Fig. 7b). Hand muscles with an MEP⁺ at baseline but with low MEP amplitude are less likely to regain strength (*p* < 0.0001, *U* = 5329; Fig. 7c). Changes in muscle strength are accompanied by gains in MEP amplitude throughout the natural

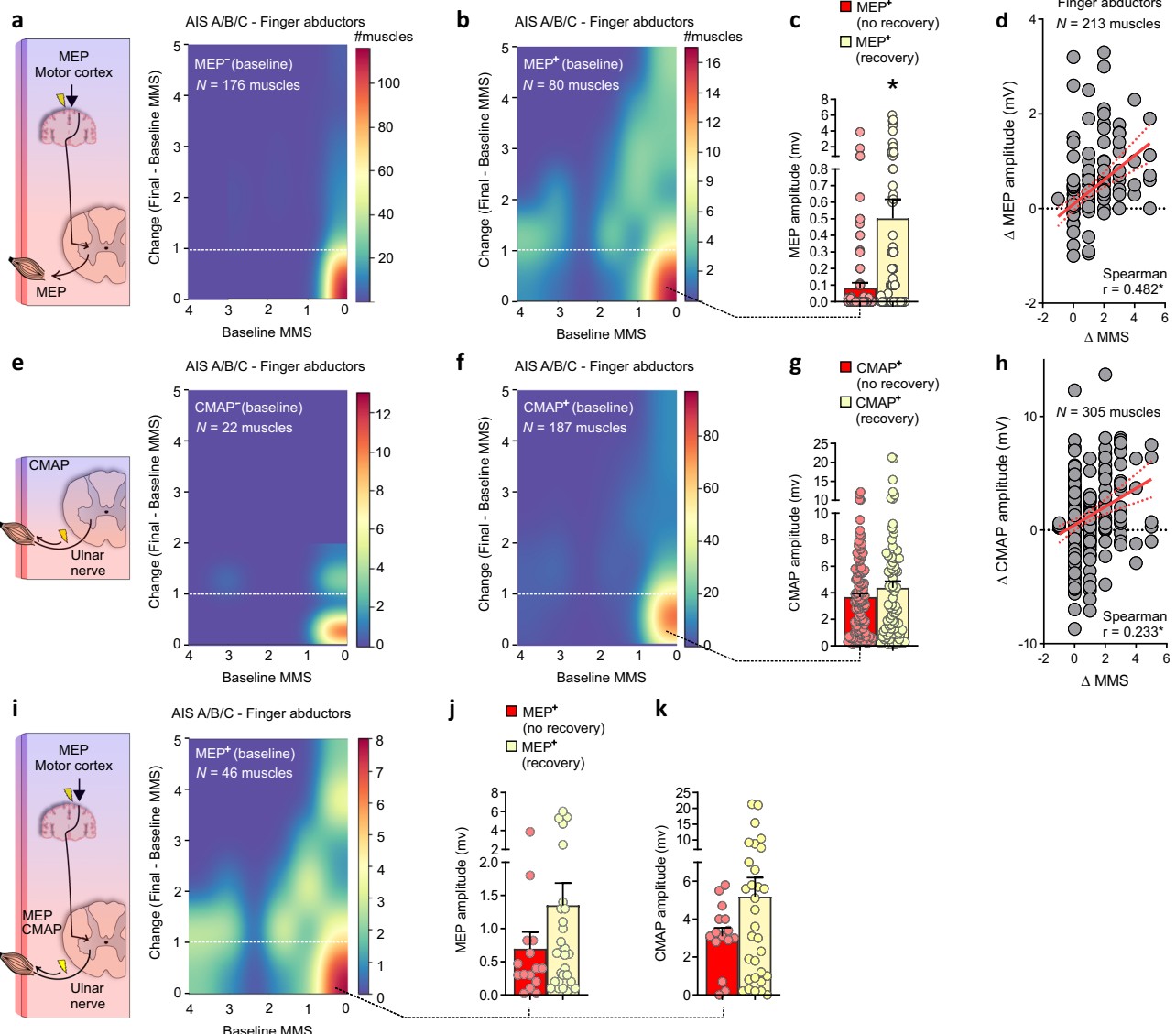

**Fig. 7 | Corticospinal tract (CST) and LMN integrity in the *abductor digiti minimi* muscle. a** In AIS A/B/C, finger abductor muscles with an MEP⁻ display limited motor recovery 1-year after SCI compared to (**b**) muscles with the presence of an MEP (MEP⁺) (yellow, above the hashed white line), including muscles with absent MMS at baseline (rightmost part of the heatmap). **c** The binary classification of MEP⁺ muscles that shows strength recovery or not was used to cluster the baseline MEP amplitude. This analysis indicates that higher MEP amplitudes were associated with increased strength recovery ($p < 0.0001$). **d** The strength recovery of finger abductors (Δ MMS) is accompanied by changes in MEP amplitude (Δ MEP amplitude) throughout the natural recovery process. **e–h** Conversely to MEP, the presence of CMAP at baseline is not strongly associated with motor recovery of the finger abductor muscles 1-year post-SCI in AIS A/B/C. **i–k** In a subgroup of individuals where both MEP and NCS studies were conducted, it is evident that muscles with an MEP⁺ at baseline and with strength recovery 1 year after SCI also show greater CMAP intensities at baseline. Data are Mean + SEM in (**c, g, j, k**) to improve visualization. In (**c, g, j–k**) the plotted amplitudes are related to the sample sizes described in (**b, f, i**) respectively. N = number of biological samples (please note that muscles from the left and right sides of the same individual are pooled in this analysis). *$p < 0.05$, Mann–Whitney tests (two-sided) in (**c, g, j, k**) Spearman correlations in (**d, h**). MMS = Muscle Motor Score, AIS = American Spinal Cord Injury Association Impairment Scale, CMAP = Compound Muscle Action Potential, MEP = Motor Evoked Potential. Outliers were left out of (**d**) (5 data points), (**h**) (2 data points) to improve visualization.

recovery process ($p < 0.0001$, $r = 0.482$; Fig. 7d). Although changes in hand muscles strength are also accompanied by an increase of CMAP throughout the natural recovery process ($p < 0.0001$, $r = 0.233$), there is a weak association between CMAP at baseline and strength recovery 1-year after SCI (Fig. 7e–k).

## CST and LMN integrity indicates impairment and recovery after SCI

Damage to the spinal cord results in muscle weakness below the lesion, with pronounced effects on hand muscles. The residual spinal cord function may be measured by the residual strength of muscles or the CST and LMN integrity (assessed by the MEP) (Fig. 8a). Individuals with absent MEP at baseline (MEP⁻) lack CST or LMN integrity and display greater initial impairment with limited strength recovery 1-year post-SCI, as measured by the total motor score ($p = 0.492$; DFn, DFd = 1, 88; $F = 0.492$). Baseline CST and LMN integrity (MEP⁺) supported motor recovery at variable degrees ($p < 0.0001$; DFn, DFd = 1, 109; $F = 16.63$; Fig. 8b). Previous work on proportional recovery in stroke indicated that individuals with MEP⁺ recover about 70% of lost upper extremity Fugl-Meyer score. Here, we show that individuals living with SCI with MEP⁺ recover to a lesser extent, ≈22% of the total motor score with greater variability compared to stroke (i.e., $R^2 = 0.132$). Additional

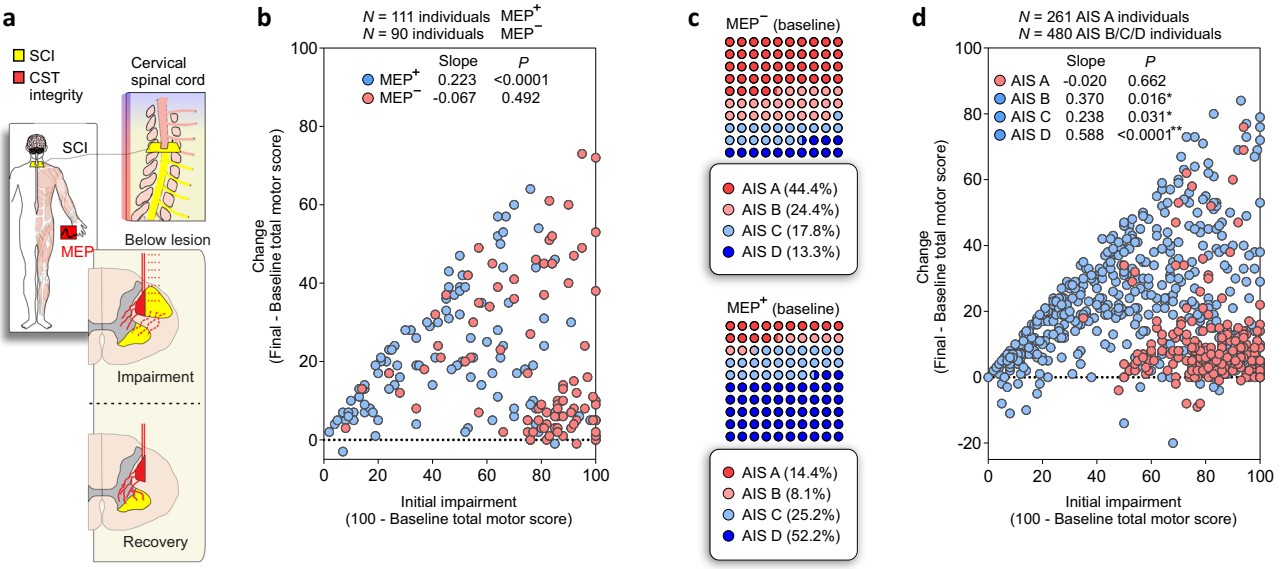

**Fig. 8 | CST and LMN integrity (as assessed by MEP) indicates impairment and recovery after SCI: individuals classified as AIS A and impaired spinal cord function (CST and LMN integrity) display limited motor recovery. a** Cervical SCI (yellow) may damage spinal cord structures and affect the spinal cord functionality below the level of injury with respective weakness of the innervated muscles. Volitional strength and strength recovery are dependent on the residual spinal cord function (red), here quantified by the residual muscle strength and MEP amplitude. **b** Individuals with absent MEP (MEP⁻) display greater damage to the descending pathway (evidenced by the greater initial impairment) and limited recovery of motor function of the spinal cord ($p = 0.492$). The presence of an MEP (MEP⁺) indicates variable levels of spinal cord or LMN damage and recovery ($p < 0.0001$). **c** Individuals with an MEP⁻ at baseline were predominantly classified as AIS A, and individuals with an MEP⁺ predominantly classified as AIS D. **d** Strength recovery

from baseline (4 weeks) to 48 weeks after SCI is shown as change in the total motor score of the ISNCSCI. For individuals classified as AIS B/C/D (blue circles), recovery is proportional to the available improvement. In AIS D, the regression represents the relationship between available ($x$) and actual ($y$) improvement ($y = 0·59x$, $p < 0.0001$). No relationship exists between available and actual improvement for sensorimotor complete lesions (AIS A, $p = 0.662$; red circles). *$p < 0.05$ simple linear regression (slope is significantly non-zero). Two individuals were excluded from the total MEP sample (203) because of absent AIS classification at baseline in (**b**). Seven individuals were excluded from the total sample (748) because of absent AIS classification at baseline in (**d**). SCI = Spinal Cord Injury, CST = Corticospinal Tract, AIS = American Spinal Cord Injury Association Impairment Scale, MEP = Motor Evoked Potential.

analysis indicated that the goodness of fit of the model increases when considering the impairment and recovery of the upper extremity only (UEMS; Supplementary Fig. 8). This analysis indicates that individuals living with SCI with MEP⁺ recover about 45% of what they have lost in the upper extremity ($p < 0.0001$; DFn, DFd = 1, 109; $F = 80.73$; Supplementary Fig. 8b); this model also shows less variability compared to the full body model ($R^2 = 0.425$). Nonetheless, it is important to highlight the great variability seen around the regression lines and note that there is no clear separation into "fitters" and "non-fitters" - which is usually observed in stroke patients. Most individuals with MEPs⁻ were classified as AIS A at baseline, on the other hand, as AIS D if an MEP⁺ was evident at baseline (Fig. 8c). Indeed, individuals classified as AIS B/C/D show recovery proportional to the largest possible improvement (AIS B: $p = 0.016$; DFn, DFd = 1, 82; $F = 6.070$; $R^2 = 0.069$; AIS C: $p = 0.031$; DFn, DFd = 1, 153; $F = 4.750$; $R^2 = 0.031$; AIS D: $p < 0.0001$; DFn, DFd = 1, 239; $F = 282.6$; $R^2 = 0.542$; Fig. 8d). No relationship between the largest possible and actual improvement is evident for individuals with a sensorimotor complete lesion (AIS A: $p = 0.662$; DFn, DFd = 1, 259; $F = 0.191$; $R^2 = 0.0007$) (Supplementary Fig. 8c–f). This analysis indicated that individuals with an MEP⁻ do not show recovery proportional to the largest possible improvement and are predominantly classified as AIS A (44.4%). The bulk of these results suggests that strength recovery can be predicted using solely baseline total motor score in AIS D and reinforces the importance of MEP in predicting the recovery of muscle strength in AIS A/B/C.

## Discussion

Natural recovery after cervical SCI relates to the segmental innervation and follows a proximal-to-distal gradient in which distal muscles of the upper limb show limited and delayed strength recovery compared to

proximal muscles. In addition, the recovery of hand muscle strength depends on the severity of spinal cord damage. In more affected individuals, non-linear interactions between residual muscle strength and the amount of recovery 1-year post-SCI challenge predictions while on average there is some recovery if the initial impairment is low to mild (baseline MMS from 3 to 4). Residual baseline strength was insufficient to predict motor recovery in severely impaired individuals–especially in the hand muscles, even when additional clinical features were included in the models. Baseline electrophysiological assessments soon after the SCI provide measures of CST and LMN integrity, and increased the prediction performance in the *abductor digiti minimi*. In the hand muscles, stronger MEPs at baseline were positive indicators of recovery, and positive changes in MEP amplitude were associated with strength recovery over time. At the person-level, our results indicate that individuals living with SCI and an MEP⁺ recover ≈45% of lost upper extremity strength–a lower proportion compared to stroke ($y ≈ 0.70x$). Our findings suggest that strength recovery can be predicted using the baseline total motor score in AIS D and highlight the importance of the MEP assessment in predicting the recovery of individuals living with more severe paralysis (AIS A/B/C).

Overall, our data support the importance of CST and LMN integrity in indicating spinal cord dysfunction and recovery, here evidenced by the residual strength and MEP at baseline. Nonetheless, integrity of these descending pathways at baseline was not always related to a good motor recovery prognostic of the hand muscles. Some hand muscles with MEP⁺ at baseline did not show motor recovery 1 year after SCI [i.e., Change in MMS = 0; 31.25% of *abductor digit minimi* muscles (25/80)]. This indicates that other variables may explain the variance of the outcomes during the recovery process and neurorehabilitation

needs to be optimized for those muscles with potential to recover early after SCI [42].

Upper limb motor recovery after SCI has been extensively studied over the past decades but most of the studies did not focus on the segmental approach described here[5,7,11–13,43–45]. Greater specificity in understanding the recovery of upper limb muscles is desirable for detecting subtle changes, because even small gains in upper limb function can have important repercussions on independence and quality of life[8,46]. Here, we expand the natural recovery to the muscle level and provide evidence of limited and delayed recovery of distal compared to proximal upper limb muscles. The greater impact on hand muscles after controlling for distance from the lesion may be explained by the amount of CST projections to spinal motoneurons, which are greater in distal muscles compared to proximal muscles[20–22]. Our data also indicates a low prevalence of LMN damage in motoneuron pools innervating the distal hand muscles (10.5% of *abductor digit minimi* muscles in AIS A/B/C showed a CMAP amplitude of 0 mV at the 4-week timepoint), supporting the idea of UMN lesions primarily accounting for our results (Table 1). Nonetheless, the assessments available in our dataset did not provide a complete assessment of LMN function for all muscles, and therefore the inability to fully account for LMN damage in the predictive models is a limitation of our study.

We hypothesize the residual CST projections to distal muscles to be the major player in muscle-specific impairment and recovery of volitional movements. This is supported by the importance of residual muscle strength (AIS D) and MEP (AIS A/B/C) for predicting strength recovery of hand muscles. Interestingly, the strength recovery of hand muscles was also associated with stronger MEP amplitudes at baseline and throughout the natural recovery process. This increase in MEP amplitude with recovery may reflect extensive spontaneous plasticity of CST projections, previously evidenced in a primate model of SCI[47]. In addition, although we acknowledge the debate around the expected electrophysiological correlate of remyelination in SCI[48], previous studies in rodents have shown the increase in MEP amplitude with oligodendrocyte precursor cells improving axonal myelination[49]. Our findings are also supported by previous clinical findings indicating that greater MEPs at baseline are associated with increased recovery of the MEP during the first year after SCI[23]. This plasticity of the residual CST projections may involve transsynaptic mechanisms rather than sprouting of spinal cord axons, which is not well observed in preclinical studies. Importantly, here, the addition of MEP as a feature increased the performance of the predictive models of strength recovery for the hand muscles. Thereby, given the high priority in regaining hand function in tetraplegia[2], patients would benefit from additional electrophysiological assessments early after the SCI. Indeed, if this residual CST functionality goes unnoticed early after the injury, the opportunity to strengthen these projections may be lost[50,51]–likely explaining why some participants displayed positive signs of spinal cord function integrity at baseline (MEP⁺) but displayed absent recovery with time. On the other hand, the great variability in our person-level models–in terms of individuals showing MEP⁻ at baseline but substantial recovery, may be explained by the persistence of spinal shock and/or the resolution of other forms of acute complications resulting from the SCI. Of note, we cannot fully account for the role of other descending spinal tracts, which have differential effects on proximal and distal upper limb control[52–54]. For example, in SCI, it is known that the reticulospinal tract assists hand control during gross finger manipulations[55]. Likewise, motor unit plasticity at the muscle level may also play a role in MEP amplitude changes. These are limitations of our study and prevent definitive conclusions about the role of the CST in our results.

Notwithstanding the greater amount of CST projections to spinal motoneurons in distal muscles compared to proximal muscles[20–22], the lack of proximal-to-distal gradient in the number of efferents that leave the spinal cord suggests that only a few motor units control fine hand movements[33]. There are over 20 intrinsic muscles in the hand[56] responsible for fine control of several degrees of freedom, which are innervated by only ≈1700 motoneurons[33] controlled by an immense neural network in primary motor areas of the cortex[18,32]. Therefore, it has been suggested that dexterous control over multiple degrees of freedom is not achieved by a finer recruitment of motor neurons in hand muscles compared to larger muscles with much grosser actions[33]. In other words, dexterous hand movements are supported by a great amount of CST projections but by much fewer motor units than previously anticipated, making any residual CST projections to motoneurons innervating the hand muscles of paramount importance in SCI. Here, the pronounced effect of the SCI on hand muscles may reflect this reliance on CST projections, indeed evidenced by the importance of residual muscle strength and MEP in predicting strength recovery of intrinsic hand muscles. This is also aligned with recent findings in non-human primates indicating lack of somatotopy in the organization of the descending CST fibers, suggesting that any lesion to the CST would have a more pronounced effects on hand/arm muscles because of the dependency on CST projections[57]. Finally, the findings of the present study may help advancing the understanding of central cord syndrome and its relation with CST integrity measured by the MEP.

The addition of SSEP did not significantly increase the performance of the prediction of strength recovery for the hand muscles. Given the predominance of sensory axons with respect to motor axons in the mixed peripheral nerves, and the increase of this ratio when moving from proximal to distal upper limb muscles[33], it is reasonable to think of the sensory information as paramount to normal hand function. We suggest that the sensory component is less important to strength recovery than it is to function. The enhanced performance of models predicting function in SCI when incorporating SSEP[4] and the reduced ability to control fine hand movements in the absence of touch and proprioceptive sensory input[58] support this conclusion. This finding mirrors the dissociation between strength and control in finger flexors in stroke, where two separate systems are responsible for poststroke hand recovery–with one system contributing mostly to the strength recovery and the other to digit individuation[59]. Here, it is reasonable to think of the MEP as a contributor to the strength system, with moderate predictive value, and MEP and SSEP to the motor control system–supported by the above-mentioned enhanced performance of models predicting function in SCI incorporating MEP and SSEP[4]. Nonetheless, we are unable to discuss the latter in detail because the present study focused on strength recovery. Future studies should further investigate the predictive value of electrophysiological assessments in the recovery of hand dexterity after SCI– not constrained to muscle strength but muscle synergies for complex motor tasks[60].

This study explored proportional recovery in SCI under the well-established framework developed for stroke[28–30,61–68]. Besides the obvious differences between these lesions and the clinical assessment scales employed in stroke (Fugl-Meyer) and SCI (ISNCSCI), a common aspect is the importance of the damage to the CST. The lack of proportionality in strength recovery and prevalence of MEP⁻ in the hand muscles suggest that individuals classified as AIS A more commonly lack CST integrity ('non-fitters' to the proportional rule)–with greater upper limb motor impairments associated with non-linearities in the strength recovery profile. Proportionality was somewhat evident for the proximal muscles in people classified as AIS A/B/C, but the strength recovery of distal muscles was hard to predict. Indeed, hand muscles were the most important players in breaking the proportionality for those individuals. The break or inversion of proportionality when the residual strength is low, or an MEP is absent, reflects the importance of preserved CST projections for motor recovery after SCI. The similarities between the UEMS of AIS A and B at 1–4 weeks but the greater recovery of AIS B at 12–48 weeks after SCI also supports the importance of CST and LMN integrity for upper extremity motor recovery.

The fact that individuals with AIS B show proportional recovery in their total motor score ($y = 0.37x$, Fig. 8D) with a respective lower prevalence of MEP$^-$ at baseline compared to AIS A (AIS A: 44.4%; AIS B: 24.4%) suggests a role for preserved CST projections and LMN integrity for upper extremity recovery even in AIS B. In individuals with AIS D, we suggest that the relationship between residual strength and strength recovery exists, and the prediction of motor recovery is possible using only the baseline MMS ($y = 0.59x$). This variable recovery in AIS-based subgroups agrees with recent findings also indicating that different subgroups present distinct recovery profiles in stroke[69]. In addition, here, the fact that ≈14% of individuals displaying a MEP$^+$ were sensorimotor complete may explain the distinct trajectory-based recovery profiles observed for this group[40]. Because it is thought that the proportional recovery from motor impairment reflects a ubiquitous neurobiological process, likely related to the biological limits of structural and functional plasticity[29], these non-linearities may indicate the constraints for recovery. Indeed, given the importance of residual strength, CST projections, and LMN integrity for proportional strength recovery in SCI, we suggest these are the basis for the strength recovery that occurs at different proportionality slopes. Finally, in individuals with MEP$^+$, the limited upper extremity recovery in SCI ($≈45\%$) compared to stroke ($≈70\%$) may reflect the direct lesion to the CST in SCI, and the segmental recovery that often occurs at a limited segmental range in SCI (i.e., ZPP)[3]. This is supported by our results indicating greater motor recovery and lower variability of the model assessing impairment and recovery in the upper extremity ($y = 0.452x$, $R^2 = 0.425$), compared to the upper/lower extremity model ($y = 0.223x$, $R^2 = 0.132$). This discussion concerning two clinical populations serves to highlight the importance of CST sparing in the context of upper limb neurorecovery, with the segmental organization and lack of damage to brain regions in SCI providing a valuable model to highlight the role of CST projections.

An important discussion concerns the methods used to calculate proportional motor recovery, and if timing and dosing of rehabilitation may contribute to the observed motor recovery. Here, we explored the more general approach of regressing change against baseline scores and MEPs as a biomarker—under the classical definition of proportional recovery in stroke to discern fitters and non-fitters[28,29]. Our results indicate that this approach yields much more variable results compared to stroke without any clear separation into fitters and non-fitters in SCI. It is important to reflect on the differences between stroke and SCI, such as LMN dysfunction[70], which adds complexity to the lesion—affecting different segments of the spinal cord (depending on the level of SCI); and the clinical assessment scales used for stroke (Fugl-Meyer motor function score: movement, coordination, and reflex) and SCI (ISNCSCI motor score: movement and strength). These differences should be explored in future studies with a direct comparison between these two populations and more sophisticated approaches to quantify proportional recovery. The proportional recovery framework remains of biological and predictive relevance in the stroke field[71] and, similar to stroke[68], is an exciting new venue for predictive studies and clinical trial design in SCI. Finally, here, similar to the stroke field[68,72–74], we attribute our findings to spontaneous biological recovery. Nonetheless, such a mechanism does not imply that rehabilitative interventions are unable to further enhance patient outcomes. In the present study, we were unable to quantify the dose and timing of therapy delivered to each individual because of the nature of the dataset. Future clinical trials should explore how interventions may induce changes that are different or greater than those expected under the proportional recovery[71].

An important question that arises from this analysis is how to optimize the strength recovery of different muscles. The answer to this question is multifaceted but it is surely an important step to understand that different muscles may follow distinct recovery profiles and the predictive value of baseline assessments. The adjunct of electrophysiological measures of volitional activity (EMG) or CST and LMN integrity (MEP) is of utmost importance for weaker muscles—especially for those with absent MMS at baseline. In future studies, identifying the potential to recover strength may help to tailor rehabilitation to novel and intensive approaches, for example, anti-NOGO therapy[75] to release the brakes of plasticity in the spinal cord and promote axonal sprouting, and paired associative stimulation[76]—to induce long-term plasticity in the CST projections. The identification of muscles with the potential to recover early after the SCI will allow the administration of novel and promising therapies during the optimal time window for recovery. The enrollment in such rehabilitation programs must also be combined with intense rehabilitation to optimize recovery[77] and avoid aberrant plasticity[78].

There is a continuous effort in improving clinical trial design and outcomes in SCI[79]. The most common method to quantifying the effects of treatment is by adding up several ordinal endpoints to form a single overall score (e.g., UEMS), but this may mislead associations and reduce statistical power[80]. There is a compelling interest in statistical models specifically designed for the analysis of complex ordinal endpoints, such as autoregressive transitional ordinal models[80]. Here, we employed machine learning methods using non-linear regressions and classification to account for the complexity of the ordinal outcomes. Future clinical trials should also consider the use of baseline-adjusted models, where the stratification based on baseline variables would improve the analysis of complex trial designs[81]. Here we show how the baseline MMS and MEP are important in predicting recovery, especially for distal hand muscles. In a limitations section, we discuss the advantages, limitations, and future directions of the muscle-specific approach applied here—including the limitations in the EMSCI dataset in terms of tracking the timing and dosing of rehabilitative training (Supplementary material).

Here, we investigated segmental strength recovery after cervical SCI and developed predictive models to evaluate the contributing factors. We demonstrated that recovery profiles of ISNCSCI key muscles are dependent on the distance from the lesion, baseline muscle strength, and SCI severity assessed by the AIS. These conclusions were possible because of the segmental analysis employed in the present study, which constitutes an alternative method to quantify recovery rates in future clinical trials. In addition, we show that lower recovery is present in the distal compared to proximal upper extremity muscles. Specifically the prediction of hand muscles strength benefited from the addition of MEP as a proxy of spinal cord function, i.e., CST and LMN integrity. While muscle strength recovery in AIS D resembled the proportional recovery seen in fitters to the proportional recovery rule in stroke, further studies should employ more sophisticated approaches to quantify proportional recovery in this population. Here, we show that individuals living with SCI with CST sparing recovered about ≈22–45% of what they have lost—a proportion much lower compared to stroke. In our predictive models, it was also evident that the CST sparing information is needed to improve prediction accuracy for more severe paralysis (AIS A/B/C), resembling predictive models in stroke in which MEP information is not needed to predict upper limb recovery for patients with mild initial impairment[82,83]. Novel interventions interacting with the neurobiological mechanisms of recovery are warranted for individuals with a potential for recovery. To further determine the therapeutic consequences, measures of CST sparing should be integrated into clinical assessment strategies following cervical SCI, especially for severe paralysis.

# Methods

## Study design

This study conducted a retrospective analysis of data from the European Multicenter Study about Spinal Cord Injury (EMSCI; ClinicalTrials.gov Identifier: NCT01571531) investigating the natural recovery after SCI. At the time of inquiry to the EMSCI (November 25th, 2020),

the database included 5794 individuals with SCI–who were assessed between 2004 and 2020. The inclusion criteria of EMSCI are: (1) single event traumatic or ischemic para- or tetraplegia, (2) first assessment possible within the first 4 weeks after incidence, (3) patient capable and willing of giving informed consent. The inclusion criteria for this study were: cervical SCI (C1–C8) and complete ISNCSCI assessments at 4, 12, 24, and 48 weeks post-SCI. Data from 749 participants with cervical SCI were analyzed, which were obtained in dedicated SCI centres: the Hohe Warte Bayreuth (Bayreuth, Germany), BG-Trauma Center (Murnau, Germany), RKU Universitäts- und Rehabilitationskliniken Ulm (Ulm, Germany), Spinal Cord Injury Center of Heidelberg University Hospital (Heidelberg, Germany), and Spinal Cord Injury Center - Balgrist University Hospital (Zurich, Switzerland). The research followed the Declaration of Helsinki and was approved by the Institutional Review Board of the abovementioned institutions: Bayrische Landesärztekammer, Ethik-Kommission (REB #188/2003; Bayreuth, Germany), Ethik-Kommission der Bayerischen Landesärztekammer (REB approval was waived because the project was treated as a data registry, but informed consent was obtained from all participants; Murnau, Germany), Universität Ulm Ethikkommission (REB #71/2005; Ulm, Germany), Universität Heidelberg Ethikkommission der Med. Fakultät (REB #S-188/2003; Heidelberg, Germany), Kanton Zürich Kantonale Ethikkommission (REB #EK-03/2004/PB_2016-00293; Zurich, Switzerland). Fifty-one individuals were excluded because of incomplete ISNCSCI assessments at baseline, as such, data from 748 individuals were analyzed (Fig. 1).

All participants were assessed around the first 4 weeks (Mean = 31 days; SD = 6.8 days) after SCI and re-assessed at 12 (Mean = 84.6 days; SD = 8.5 days), 24 (Mean = 168.5 days; SD = 11.1 days), and 48 weeks (Mean = 356.7 days; SD = 54 days). A subset of participants ($N = 440$) was additionally assessed in the very acute phase of SCI (Mean = 8.7 days; SD = 4.6 days), which was specifically used to describe motor recovery (not for predictive modelling, considering the potential effects of spinal shock and the fact that not all patients are able to be assessed at very early timepoints). Subsets of participants also underwent electrophysiological multimodal assessments of motor evoked potentials (MEPs; $N = 203$), somatosensory evoked potentials (SSEPs; $N = 313$), and nerve conduction studies (NCS; $N = 280$) of the nerves of the upper limb innervating hand muscles. The research followed the Declaration of Helsinki and was approved by the Institutional Review Board of the above-mentioned institutions. Supplementary Table 1 shows demographic and clinical characteristics of the research participants.

Muscle strength was measured according to ISNCSCI in ten key muscles for each side of the body for each participant: five upper limb muscles [elbow flexors (C5), wrist extensors (C6), elbow extensors (C7), finger flexors (C7), and finger abductors (T1)] and five lower limb muscles [hip flexors (L2), knee extensors (L3), ankle dorsiflexors (L4), long toe extensors (L5) and ankle plantar flexors (S1)]. Each muscle was scored from 0 to 5 (Muscle Motor Score; MMS) following the recommendations of the ISNCSCI[31,84]. Thus, the total motor score has a maximum of 50 per side, and 100 per person. The upper extremity motor score (UEMS) consists of a maximum of 25 per side and 50 per person.

### Segmental strength recovery of upper limb muscles after SCI
Non-parametric statistics and non-linear regression using random forest regressors were used to explore segmental strength recovery after SCI. The distance (DST) in myotomes between the ISNCSCI motor level and the muscle myotome on each side of the body was used to split the dataset, in order to control for the distance from the motor level when comparing muscles (negative DST values denote myotomes caudal to the motor level). It is known that the recovery of motor function in spinal segments below the ISNCSCI motor level will typically occur approximately one to three levels caudal to it (DSTs −1 to −3)[3]. Based on this information, our analysis encompassed muscles 1–4 levels below

the motor level (DSTs −1 to −4). Muscles from the left and right sides of the same participant were considered as independent samples after correcting for DST. For the random forest non-linear regression and classification models, muscles with a baseline strength of 5 were excluded to control for ceiling effects.

### Prediction of segmental strength recovery
The residual strength after SCI is indicative of preserved supraspinal connections to the muscles. To test if strength recovery is related to the amount of residual strength at baseline, we assessed the recovery using the proportional recovery framework previously employed to describe stroke recovery[28–30,61–68]. The change of MMS between baseline (4 weeks) and endpoint (48 weeks) were regressed against the initial impairment (5 - Baseline MMS), the initially preserved motor function, to predict motor recovery in relation to the initial impairment.

In addition to the baseline MMS, we explored the inclusion of variables extracted from the sensory components of the ISNCSCI as additional features in the machine learning models[31]. The codes utilized for data curation and analysis are provided in Supplementary Software 1[85]. The light touch (LT) and pin prick (PP) sensation scores of the dermatome corresponding to each myotome of the upper extremity were analyzed. In addition to LT and PP scores, the DST was also considered as a feature in the machine learning models. General models included all AIS grades and muscles (features used in model 1: AIS, MMS, DST, LT, and PP). Muscle identity (i.e., key muscles in the ISNCSCI: C5/elbow flexors, C6/wrist extensors, C7/elbow extensors, C8/finger flexors, or T1/finger abductors) was taken into account in a subsequent step (features used in model 1 + Muscle identity: AIS, Muscle, MMS, DST, LT, and PP). We also created muscle-specific models by using data from each individual key muscle (models 2–6) rather than pooled data from all muscles. These models were important to understand how the strength recovery prediction differed between muscles. Random forest classifiers were used to predict segmental strength recovery after SCI: model 2 (elbow flexors), model 3 (wrist extensors), model 4 (elbow extensors), model 5 (finger flexors), and model 6 (finger abductors).

MMS alone may not reflect CST and LMN sparing[86] and, therefore, we explored the predictive value of electrophysiological multimodal assessments in a subsample of the participants. This subsample consisted of individuals classified as AIS A/B/C that had undergone electrophysiological multimodal assessments at baseline [within the first 4 weeks (Mean = 31 days; SD = 6.8 days)]. The assessments conducted were MEPs on the *abductor digiti minimi*, SSEPs from the ulnar nerve, and NCS of the ulnar nerve. A detailed description of the materials and methods used in the electrophysiological multimodal assessments of the hand muscles is presented in the Supplementary Material[4,23,27].

### Electrophysiological multimodal assessments scoring system.
Transformation to a scoring system was guided by clinical normative values[4]. All neurophysiological examinations were rated as normal (2 points), impaired (1 point), or abolished (0 points) as shown in Supplementary Table 2. The scoring system resulted in an ordinal value with a maximum of 3 points for MEP (MEP score), 3 points for SSEP (SSEP score), and 3 points for NCS (NCS score).

### Time-course of motor evoked and compound muscle action potentials recovery
For some participants, follow-up assessments of MEP were performed. The resulting time-course of MEP amplitude recovery provides an electrophysiological perspective on motor recovery. Muscles at and up to eight segments caudal to the motor level of SCI were considered for the analysis of the time-course of MEP recovery. Muscles with a baseline MMS of 5 were excluded to reduce ceiling effects. The follow-up assessment was conducted at different time points. A total of 259 *abductor digiti minimi* muscles MEPs were available at baseline, but in

46 only the baseline MEP assessment was performed and those were not included in the analysis. A total of 213 MEPs were included in this analysis, of which 133 had the endpoint MEP assessment conducted at 48 weeks post-SCI, 16 at 24 weeks, and 64 at 12 weeks. A total of 332 *abductor digiti minimi* muscles CMAPs were available at baseline, but in 27 muscles only the baseline CMAP assessment was performed and those were not included in the analysis. A total of 305 CMAPs were included in this analysis, of which 196 had the endpoint CMAP assessment conducted at 48 weeks post-SCI, 19 at 24 weeks, and 90 at 12 weeks.

## Statistical analysis

The EMSCI study is designed as a European, multicenter, observational study of the natural course of neurological, functional, independence, and pain parameters in SCI patients. Currently, the sample size of the EMSCI study is above the estimated sample size of 5500 individuals (ClinicalTrials.gov Identifier: NCT01571531). In the cohort of individuals described in this study, we based our sample size estimation on previous studies detecting the effects of natural recovery in SCI at the muscle level (167 individuals[13]) and describing the proportional recovery rule in stroke (41 individuals[62], 93 individuals[28], 385 individuals[63]). The sample size utilized in the present study (748 individuals) is, thus, above the sample size used by similar studies in the past, which were able to detect the expected effects of natural or spontaneous recovery. Furthermore, this is the first study to assess the role of CST integrity in the proportional recovery after SCI - analyzing MEPs in a sample of 203 individuals (also above the 93 individuals utilized to describe this effect in stroke[28]). Statistical significance was set at $\alpha = 0.05$. Data normality was assessed and indicated the use of parametric or non-parametric statistics. The analysis was conducted using Python scikit-learn (machine learning analysis and data visualization), SPSS® Statistics (descriptive analysis, median comparisons), Excel (data sorting), LabVIEW® (data visualization and sorting), and GraphPad Prism® (data visualization and descriptive analysis).

Data normality was assessed using the Shapiro–Wilk test. The unit of measure was the individual or each muscle and data were expressed using median (Fig. 3a) or median, interquartile intervals, and 5–95 percentiles (Supplementary Figs. 2–5)—unless otherwise noted to improve visualization (Mean ± SD or Mean ± SEM). The McNemar's test was used to compare muscle strength recovery (%muscles with MMS ≥ 3) over time (Fig. 3d–h) and the accuracy of the random forest classifiers (Fig. 6f). The Kruskal–Wallis H test was used as a non-parametric alternative to the one-way ANOVA to determine if there were statistically significant differences between the strength of distinct ISNCSCI key muscles, adjusted using the Dunn's multiple comparisons correction (Supplementary Figs. 2–5). The Mann–Whitney U test was used to compare differences between two independent groups (Fig. 7c, g, j, k). Multiple Mann–Whitney tests were also used to compare ranks and multiple comparison adjustments were performed using the false discovery rate and the two-stage step-up method of Benjamini, Krieger, and Yekutieli (Fig. 3a, b)[87,88]. Spearman correlation was used to test the relation between non-parametrical variables (Fig. 7d, h). Simple linear regressions were used to determine the proportional recovery after SCI and test the if the slope was significantly non-zero (Fig. 8b, d).

To assess the predictive value of baseline MMS, we followed recent recommendations from studies on proportional recovery after stroke and performed descriptive statistics of strength recovery data[67], implemented machine learning approaches[64], controlled for ceiling effects[63], and performed non-linear regression models using decision trees[63,89]. Random forest regressors were conducted using 50% of the dataset for training and 50% for testing with 100 trees (estimators). The random forest algorithm fitted several classifying decision trees on various sub-samples of the dataset and used averaging to improve the predictive accuracy and control over-fitting (in contrast to the

original method)[90]. The non-linear regression model fit was assessed by the $R^2$ and prediction error [average of abs(predicted-true)] and qualitatively by visual inspection of the regression lines. To understand the effects of the dependency in the dataset, which contained muscles from the left and right sides of the same participant, we performed a dataset split and applied the random forest regressors as described above.

Also following recent recommendations[63,64], we explored supervised machine learning models using random forest classifiers with additional baseline predictors – including quantitative multimodal electrophysiological assessment. The classifier was trained to predict motor recovery based on the change of MMS score between baseline and 48 weeks post-SCI ('Recovery': an increase of at least 1; 'No recovery': no change or decline). The following baseline features were used as predictors in the models (included features varied across models, as specified in relevant portions of the results): AIS, MMS, DST, LT, PP, Muscle identity, MEP amplitude, MEP score, SSEP amplitude, SSEP score, CMAP amplitude, F-wave persistence, and NCS score. Random forest classifiers were constructed using 100 trees (estimators). Evaluation was carried out using leave-one-muscle-out cross-validation. The performance of the random forest classifiers was assessed by the precision, recall/sensitivity, specificity, and F1-score of the predictions. In addition, receiver operating characteristic and precision-recall curves[91] were used to obtain the area under the curve (ROC AUC and PR AUC, respectively)—indicating the overall performance of the models as additional elements are added. The differences between the models including the electrophysiological features and models based only on clinical features were assessed using the McNemar's test. To understand the importance of each feature to the prediction, we also reported the feature importance calculated as the decrease in node impurity weighted by the probability of reaching that node (computed using 50% of the dataset for training and 50% for testing; Supplementary Table 3). The node probability was calculated by the number of samples that reach the node, divided by the total number of samples. The higher the feature importance value the more important the feature. Finally, to understand the effects of the dependency in the dataset, which contained muscles from the left and right sides of the same participant, we performed an additional cross-validation procedure. We employed the leave-one-subject-out cross-validation to ensure that none of the data from the test participant was used to train the models.

Spearman correlation was used to explore the relationship between the change in MMS with change in MEP and CMAP amplitude. The relation between impairment and recovery of the spinal cord function was assessed using the motor component of the ISNCSCI, MEP amplitudes, and linear regression models. The ISNCSCI is a Likert-like scale, and thus is a summary of multiple Likert-like items, comprising ordinal data[92]. We considered that the combination of multiple items renders the parametric statistical approaches applied here feasible[63]. Mathematical coupling is an important statistical consideration when regressing the initially preserved motor functions against change scores (change in MMS from baseline to endpoint), which was extensively debated over the past years in the stroke recovery field[63–65,67]. Clustering algorithms commonly used in proportional recovery studies can bias the regression toward high values because of the low variability of the clustered data at the endpoint (after mathematically removing non-fitters)[63–65]. To address these issues, we refrain from using mathematical clustering, instead, we clustered the dataset based on physiological biomarkers[28,29,61,66,68], i.e., the severity of SCI (AIS classification)[31] or the presence(MEP+)/absence(MEP−) of MEP[28].

## Reporting summary

Further information on research design is available in the Nature Portfolio Reporting Summary linked to this article.

## Data availability

This study conducted a secondary analysis of the European Multicenter Study about Spinal Cord Injury (EMSCI; NCT01571531) in accordance with the terms agreed upon the receipt of the dataset (REB #20-5914 – University Health Network). The deidentified participant data utilized in this study have been deposited in the SYNAPSE database under public access [https://doi.org/10.7303/syn50900334]. Additional data are available under restricted access and can be obtained by request to the EMSCI. Source data are provided with this paper.

## Code availability

Source codes are provided with this paper (https://doi.org/10.5281/zenodo.7545219) [85].

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

## Acknowledgements

We would like to thank all the research volunteers and the EMSCI study group. This work was supported by the Wings for Life Spinal Cord Research Foundation (Project #210; J.Z.), the Nogo Inhibition in Spinal Cord Injury Project (NISCI - HZ2020; A.C.), and the Canadian Institutes of Health Research (CIHR, MFE – 183072; G.B.).

## Author contributions

G.B. data curation and harmonization, carried out the formal analysis, interpreted and wrote the results and figures, wrote the first draft of the paper, and revised the paper. G.L. and S.K. revised the final version of the paper. R.A., D.M., Y-B.K., and N.W. acquired and analyzed the data, and revised the final version of the paper. R.R., M.S., and A.C. acquired and analyzed the data, revised and discussed the initial drafts of the paper, and revised the final version of the paper. J.Z. conceptualized the study, supervised the study, carried out the formal analysis, interpreted and wrote the results and figures, wrote the first draft of the paper, and revised the paper.

## Competing interests

The authors declare no competing interests.
