## [Peer Review File · Nature Communications]

Segmental motor recovery after cervical spinal cord injury
relates to density and integrity of corticospinal tract
projectionsREVIEWER COMMENTS

Reviewer #1 (Remarks to the Author):

This is a very thorough, important, and quite extensive piece of work. The main message seems to be that variability in recovery and difficulty of prediction increases with severity of the SCI, especially in the hand intrinsics.

I have comments but they are not really major in anyway but would make, I think, the manuscript even better.

1. The main emphasis of the manuscript is muscle strength but a great deal of work on the CST and the hand is about dexterity. The authors make no mention of this. A paper by Xu, Ejaz, Hertler et al JNP 2017, for example, shows a dissociation between strength and control in finger flexors. It would be interesting to get the authors view on this.
2. The authors in the discussion talk about the fact that MEPs can change in amplitude over time and that this relates to recovery. Can they say a little more whether they think this is related to training? Does anything in there data suggest that early training is predictive? They have so many patients it is surprising that intervention dose and intensity was not used as a predictor.
3. It would help if the authors discussed how they defined and calculated proportionality given that there has been quite some debate around this issue as of late. Can refer to Kundert, Goldsmith et al 2019 with respect to the regression approach and Goldsmith, Kitago et al 2021 (bioRxiv 2021).
4. Line 145 is a little unclear - I think they are saying when controlling for distance from lesion.
5. The sentence starting on line 449 with citation 33 should be elaborated upon, because its relation to the next sentence on line 451 is unclear.

Reviewer #2 (Remarks to the Author):

Overall, the results of this study mirror observations of people recovering from stroke. Patients with initially mild upper limb impairment tend to recover well, with recovery closely related to initial impairment. In contrast, recovery cannot be predicted from clinical assessment alone for patients with initially more severe upper limb impairment. In both groups of patients, MEP status is a useful biomarker that can identify which patients with initially severe upper limb impairment are likely to make a meaningful recovery. Also similar to stroke, improvements in MEP parameters parallel improvements in motor function during recovery. These similarities could be stated more clearly, as the big picture tends to get lost in the descriptions of patient subsets, different muscles, and distances from the lesion.

The authors have applied the concept of proportional recovery to SCI patients, and this seems to be a novel approach. The proportional recovery 'rule' regresses the change in score against initial impairment – which is calculated as maximum possible score minus baseline score. The method reported here regressed change in MMS against baseline MMS, rather than initial impairment. Please note that this differs from the proportional recovery framework referenced in this section. Baseline MMS and initial impairment are not the same thing, as the sentence beginning Line 586 seems to state. Figure 8b correctly identifies initial impairment as 100 – baseline total motor score. It's not clear how proportional recovery was evaluated for individual muscles, particularly given the range of possible scores is only 0 to 5. Previous work on proportional recovery in stroke finds that patients who are MEP+ recover about 70% of what they have lost. Anyone familiar with this previous body of work will be expecting to understand what proportion of the initial impairment is recovered in this patients – but this isn't apparent. The term "proportional" is used in an ambiguous way, and seems to mean

"some" versus "no" recovery.

Additional specific comments are provided below.

Methods, Line 550: It's not accurate to say that all participants were assessed "within" the first 4 weeks after injury if the mean time point was 31 days after injury.

Methods, Line 554: How was the additional measure at around 9 days for 440 patients "specifically used to describe motor recovery"?

Methods, Line 578: Is it appropriate to consider the two sides of the body as "independent" for statistical analysis? Clearly they are not.

Line 429: Might remyelination play a role in recovery of MEP amplitude and strength?

Line 434: The authors note the importance of obtaining MEP status early after SCI (suggest particularly for AIS A/B/C) so that residual CST functionality can be recognised and strengthened during rehabilitation. It's not clear how this study can be used to inform selection of patients into clinical trials, or tailoring of therapy in routine clinical practice. How would one make a prediction for an individual patient, and how accurate would that prediction be? Would MEP status need to be identified in all affected muscles, or just some? Is it the case that therapy would then be targeted to specific muscles, instead of to potentially useful movements?

Dear Reviewers:

Thank you for taking the time to review our manuscript and providing feedback that we agree will improve and strengthen our study. We appreciate your feedback, comments, and queries. We have included the Reviewer, Question/Query with our response. It is indicated in the rebuttal letter if the change was incorporated in the revised manuscript.

Reviewer #1 (Remarks to the Author):

This is a very thorough, important, and quite extensive piece of work. The main message seems to be that variability in recovery and difficulty of prediction increases with severity of the SCI, especially in the hand intrinsics. I have comments but they are not really major in anyway but would make, I think, the manuscript even better.

Thank you for taking the time to review our manuscript and providing feedback that we agree will improve and strengthen our study.

1. The main emphasis of the manuscript is muscle strength but a great deal of work on the CST and the hand is about dexterity. The authors make no mention of this. A paper by Xu, Ejaz, Hertler et al JNP 2017, for example, shows a dissociation between strength and control in finger flexors. It would be interesting to get the authors view on this.

Response: We have added more about this discussion on pp. 26, lines 517-526. We now offer the reader a perspective comparing what was seen in stroke [Xu, J. et al. Separable systems for recovery of finger strength and control after stroke. J. Neurophysiol. (2017)] with our findings. Because our study did not focus on the measure of hand function (hand dexterity), we seek support from a previously published article that analyzed the EMSCI dataset and found a predictive value for SSEP (in combination with the other electrophysiological assessments of MEP and NCS) in the prediction of the recovery of function using the Spinal Cord Independence Measure (SCIM). SCIM is designed for individuals with SCI and assesses performance in activities of daily living and mobility. This was a limitation of the present study, we consider that the predictive value of the electrophysiological multimodal assessments in gross independence measurements (measured using the SCIM) [Hupp M., et al. Electrophysiological Multimodal Assessments Improve Outcome Prediction in Traumatic Cervical Spinal Cord Injury. J Neurotrauma. (2018)] and the data presented in the present work are not sufficient to discuss the role of CST for hand dexterity following SCI. Thereby, we recommend that future studies should address this topic (on pp. 26, lines 524-526).

2. The authors in the discussion talk about the fact that MEPs can change in amplitude over time and that this relates to recovery. Can they say a little more whether they think this is related to training? Does anything in there data suggest that early training is predictive? They have so many patients it is surprising that intervention dose and intensity was not used as a predictor.

Response: Unfortunately, we are not able to discuss if the observed change in MEP amplitude over time is related to the dosetiming of rehabilitation. We acknowledge the limitation of our study in terms of not-measuring the dose and timing of rehabilitation (on pp. 31, lines 623-624 and in the Supplementary Material). We also discuss how this limitation may have affected our findings (on pp. 29, lines 593-598). Here, we would like to thank the reviewer for this important comment that warrants further investigation. We acknowledge that different recovery settings may not be well described by the proportional recovery described here. Future studies can validate and further investigate our findings using other datasets (e.g., NASCIS III, SyGEN, NACTN, and STASCIS). Similar to the stroke field, it is difficult to control the rehabilitation delivered to each individual within the standard of care in SCI; and it is not ethical to simply not provide any rehabilitation to certain individuals (to serve as a control, non-rehab group). These factors hamper the understanding of the concerns raised by this reviewer. Moving forward, we will address these issues in upcoming studies in rodents, where we can control for the above-mentioned aspects of rehabilitation dose and timing.

3. It would help if the authors discussed how they defined and calculated proportionality given that there has been quite some debate around this issue as of late. Can refer to Kundert, Goldsmith et al 2019 with respect to the regression approach and Goldsmith, Kitago et al 2021 (bioRxiv 2021).

Response: We have added an extensive paragraph in the Discussion to address this concern. We acknowledge the extensive debate in the stroke field and re-emphasize all the lessons learned from these previous discussions; and how we used this body of knowledge to build our analysis plan to avoid common pitfalls. In short, (1) we performed descriptive statistics of strength recovery data, (2) implemented machine learning approaches, (3) controlled for ceiling effects, (4) performed non-linear regression models using decision trees, and (5) explored how other forms of measure at baseline would affect the prediction. Please see these additions on pp. 28-29, lines 565-593.

4. Line 145 is a little unclear - I think they are saying when controlling for distance from lesion.

Response: yes, we added the word "especially" to build on the fact we are controlling for the distance from the lesion. Please see this change on pp. 7., line 152.

5. The sentence starting on line 449 with citation 33 should be elaborated upon, because its relation to the next sentence on line 451 is unclear.

Response: We have added a sentence to improve the link between the above-mentioned sentences. Please see this change on pp. 25-26, lines 498-508.

Additional comments:

Please note that during the process of re-analysis of the dataset we found a typo on the regression slope presented in Figure 8d (the slope for AIS D was 0.622, the correct slope is 0.588). Also note that we added the following references:

- Xu, J. et al. Separable systems for recovery of finger strength and control after stroke. *J. Neurophysiol.* 118, 1151–1163 (2017).
- Goldsmith, J. et al. The proportional recovery rule redux Arguments for its biological and predictive relevance. *bioRxiv* (2021).
- Cramer, S. C. Repairing the human brain after stroke: I. Mechanisms of spontaneous recovery. *Ann. Neurol.* 63, 272–287 (2008).
- Zeiler, S. R. & Krakauer, J. W. The interaction between training and plasticity in the poststroke brain. *Curr. Opin. Neurol.* 26, 609–16 (2013).
- Cassidy, J. M. & Cramer, S. C. Spontaneous and Therapeutic-Induced Mechanisms of Functional Recovery After Stroke. *Transl. Stroke Res.* 8, 33–46 (2017).
- Eisen, A. & Lemon, R. The motor deficit of ALS reflects failure to generate muscle synergies for complex motor tasks, not just muscle strength. *Neurosci. Lett.* 762, 136171 (2021).
- Schambra, H. M. et al. Differential Poststroke Motor Recovery in an Arm Versus Hand Muscle in the Absence of Motor Evoked Potentials. *Neurorehabil. Neural Repair* 33, 568–580 (2019).
- Morecraft, R. J., Stilwell-Morecraft, K. S., Ge, J., Kraskov, A. & Lemon, R. N. Lack of somatotopy among corticospinal tract fibers passing through the primate craniovertebral junction and cervical spinal cord: pathoanatomical substrate of central cord syndrome and cruciate paralysis. *J. Neurosurg.* 136, 1395–1409 (2022).

Reviewer #2 (Remarks to the Author):

Overall, the results of this study mirror observations of people recovering from stroke. Patients with initially mild upper limb impairment tend to recover well, with recovery closely related to initial impairment. In contrast, recovery cannot be predicted from clinical assessment alone for patients with initially more severe upper limb impairment. In both groups of patients, MEP status is a useful biomarker that can identify which patients with initially severe upper limb impairment are likely to make a meaningful recovery. Also similar to stroke, improvements in MEP parameters parallel improvements in motor function during recovery. These similarities could be stated more clearly, as the big picture tends to get lost in the descriptions of patient subsets, different muscles, and distances from the lesion.

Thank you for taking the time to review our manuscript and providing feedback that we agree will improve and strengthen our study.

Response: *We worked to improve and clarify the main take-home messages of our study. We reworked the title and abstract and key portions of the manuscript (the last paragraph of the introduction, the first paragraph of the discussion and the conclusion paragraph) to better reflect the main findings and also the parallel with the stroke field. Please see the changes on pp. 1 (title and abstract), on pp. 4-5, lines 103-114 (last paragraph of the introduction), on pp. 22, lines 431-438 (first paragraph of the discussion), on pp. 31-32, lines 625-645 (conclusion paragraph). We would also like to kindly state that the segmental analysis was the primary motivation for the study and necessary for us to reach the main conclusions of our study; this analysis was lacking in the SCI field and constitutes a significant contribution in our view, in addition to the person-level findings on injury severity trends and MEP biomarkers – we highlight this on pp. 29, lines 583-590.*

The authors have applied the concept of proportional recovery to SCI patients, and this seems to be a novel approach. The proportional recovery ‘rule’ regresses the change in score against initial impairment – which is calculated as maximum possible score minus baseline score. The method reported here regressed change in MMS against baseline MMS, rather than initial impairment. Please note that this differs from the proportional recovery framework referenced in this section.

Response: *We have changed the x-axis of the non-linear regression in Figure 4 from “baseline MMS” to “Initial impairment (5 – Baseline MMS)”. This should avoid confusion and provide a better alignment with the previous work on the proportional recovery rule in stroke. Now all the regressions (non-linear and linear) are displayed under the original proportional recovery framework (change scores against initial impairment). Please see these changes throughout the manuscript, especially in Figure 4 on pp. 11.*

Baseline MMS and initial impairment are not the same thing, as the sentence beginning Line 586 seems to state. Figure 8b correctly identifies initial impairment as 100 – baseline total motor score. It’s not clear how proportional recovery was evaluated for individual muscles, particularly given the range of possible scores is only 0 to 5.

Response: *In addition to the changes described in the above response [changing the x-axis of the non-linear regression in Figure 4 from “baseline MMS” to “Initial impairment (5 – Baseline MMS)”], we have changed the text in the methods accordingly. Please see this change on pp. 34, line 708.*

Previous work on proportional recovery in stroke finds that patients who are MEP+ recover about 70% of what they have lost. Anyone familiar with this previous body of work will be expecting to understand what proportion of the initial impairment is recovered in this patients – but this isn’t apparent.

Response: *We have improved the description of the proportional recovery in SCI and provided the reader with a better and clearer comparison with the previous work in stroke. These changes are indicated in the pages and lines provided in the response to the first query.*

The term “proportional” is used in an ambiguous way, and seems to mean “some” versus “no” recovery.

Response: *We have reworked the use of the term, please see these changes on pp. 10, lines 190-202 and on pp 22, line 424.*

Additional specific comments are provided below.

Methods, Line 550: It’s not accurate to say that all participants were assessed “within” the first 4 weeks after injury if the mean time point was 31 days after injury.

Response: *Please see this change on pp. 32, line 670.*

Methods, Line 554: How was the additional measure at around 9 days for 440 patients “specifically used to describe motor recovery”?

Response: *The very acute time point was used specifically to describe motor recovery. We would like to state that because of the complications from the SCI, which is often traumatic and requires decompressive surgery (among others), not all participants are capable of undergoing a clinical assessment at this very early stage. Secondly, because of spinal shock, the electrophysiological assessments may be biased at this early stage, where the spinal cord is unresponsive due to spinal shock. Thereby, we refrained from using any baseline predictor from the very acute assessment – all the predictions were conducted using baseline features extracted from the acute time point (≈4 weeks post-SCI). This allowed us to reduce the influence of spinal shock on our baseline features of muscle strength and electrophysiology while increasing the sample size available for analysis (the very acute dataset had only 440 participants in contrast with the acute dataset – which contained 749 individuals). The additional measure at around 9 days for 440 patients was, thereby, only used to describe motor recovery over time. We decided to use this dataset for describing the motor recovery to capture the full spectrum of motor recovery following the lesion, while not contaminating our predictions with baseline features that may have been affected by spinal shock. This point has been clarified in the Methods (on pp. 33, lines 674-676).*

Methods, Line 578: Is it appropriate to consider the two sides of the body as “independent” for statistical analysis? Clearly they are not.

Response: We have performed a full reanalysis of the dataset using two approaches to account for the dependency in the dataset. First, for the descriptives of motor recovery and non-linear regressions using random forest regressors, we employed a dataset split and presented data for the left and right sides of the body separately. This analysis confirmed the findings of the merged dataset and is described in detail in two novel supplementary figures (Supplementary Figures 6 and 7). Please see these changes on pp. 8, lines 165-168, on pp. 10-11, lines 208-212, on pp. 39, lines 804-806. Secondly, we conducted an additional layer of validation in our supervised machine learning models using a leave-one-subject-out cross-validation procedure. This ensures that our models were trained using a dataset that did not contain any muscle of the test subject – at each fold of the cross-validation. Although machine learning models are considered robust enough to deal with such dependency in the datasets, this additional cross-validation layer strengthens the quality of the evidence concerning the main findings of our work. Please see these additions on pp. 15, lines 291-295, on pp. 17, lines 333-339, on pp. 40, lines 830-833, and in Supplementary Tables 4 and 5. Overall, the process of re-analysis of the dataset was very important for controlling for the dependency in our measures as well as for comprehensively reviewing all the analyses conducted.

Line 429: Might remyelination play a role in recovery of MEP amplitude and strength?

Response: We added more to this discussion, please see this change on pp. 24, lines 467-470 and on pp. 24-25, lines 481-484.

Line 434: The authors note the importance of obtaining MEP status early after SCI (suggest particularly for AIS A/B/C) so that residual CST functionality can be recognised and strengthened during rehabilitation. It's not clear how this study can be used to inform selection of patients into clinical trials, or tailoring of therapy in routine clinical practice. How would one make a prediction for an individual patient, and how accurate would that prediction be? Would MEP status need to be identified in all affected muscles, or just some? Is it the case that therapy would then be targeted to specific muscles, instead of to potentially useful movements?

Response: We recognize the value of the measure of CST sparing by the MEP in more severely affected individuals (AIS A/B/C) as an important indicator of motor recovery. The clinical assessment usually discerns those without sacral sparing (motor and sensory, or motor only) – which are AIS A and B individuals, and individuals with sacral sparing (AIS C). Within these 3 subgroups, we have shown that the presence of CST sparing – assessed by the MEP, may indicate a subgroup with a better recovery prognostic of hand strength. We believe that the presence of a MEP* may be used to inform the selection of patients for clinical trials or tailoring of therapy in routine clinical practice, especially, for treatments aiming to restore hand function – a top priority in cervical SCI. The use of predictive models in this study was primarily intended to understand the factors that contribute to segmental recovery. In other words, we were interested in how predictions changed between models with different predictors, more so than in the absolute prediction accuracy achieved. Future work may focus specifically on optimizing prediction at both the segmental and person levels. Avenues to improve predictions may include more sophisticated machine learning models, alternate choices of measures and thresholds for defining recovery and, at the person level, accounting for the presence of MEPs in multiple muscles. MEPs from proximal muscles would help to detect UMN and LMN lesions to the C4, C5 segments – the spinal cord segments most directly affected by SCI. There are also methodological difficulties in measuring CMAP from proximal muscles such as biceps brachii, thereby, the MEP could help the prediction for proximal muscles. Lastly, we think that the predictive value even for a single upper limb muscle is meaningful because of the top priority in regaining hand function after cervical SCI. Considering the present focus on segmental recovery and the above-stated role of the predictive models in our analysis, we have opted to leave the important point of person-level prediction to future work.

Prediction for an individual patient: although we support the use of the MEP as an indicator of CST integrity and recommend that additional muscles are assessed, the effect of the CMAP was lost in our analysis because of selection bias. In other words, because of the prevalence of SCIs in C4, C5 segments our dataset “likely” mostly contain LMN lesions to these segments, spanning the LMNs innervating the hand muscles. Thereby, predictions for an individual patient may still benefit from the CMAP and future studies should investigate the predictive value of CMAP and MEP in a subsample of lower cervical lesions (C7-T1) or using CMAP obtained from different nerves such as the musculocutaneous nerve.

Additional comments:

Please note that during the process of re-analysis of the dataset we found a typo on the regression slope presented in Figure 8d (the slope for AIS D was 0.622, the correct slope is 0.588). Also note that we added the following references:

- Xu, J. et al. Separable systems for recovery of finger strength and control after stroke. *J. Neurophysiol.* 118, 1151–1163 (2017).
- Goldsmith, J. et al. The proportional recovery rule redux Arguments for its biological and predictive relevance. *bioRxiv* (2021).
- Cramer, S. C. Repairing the human brain after stroke: I. Mechanisms of spontaneous recovery. *Ann. Neurol.* 63, 272–287 (2008).
- Zeiler, S. R. & Krakauer, J. W. The interaction between training and plasticity in the poststroke brain. *Curr. Opin. Neurol.* 26, 609–16 (2013).
- Cassidy, J. M. & Cramer, S. C. Spontaneous and Therapeutic-Induced Mechanisms of Functional Recovery After Stroke. *Transl. Stroke Res.* 8, 33–46 (2017).
- Eisen, A. & Lemon, R. The motor deficit of ALS reflects failure to generate muscle synergies for complex motor tasks, not just muscle strength. *Neurosci. Lett.* 762, 136171 (2021).
- Schambra, H. M. et al. Differential Poststroke Motor Recovery in an Arm Versus Hand Muscle in the Absence of Motor Evoked Potentials. *Neurorehabil. Neural Repair* 33, 568–580 (2019).

-Morecraft, R. J., Stilwell-Morecraft, K. S., Ge, J., Kraskov, A. & Lemon, R. N. Lack of somatotopy among corticospinal tract fibers passing through the primate craniovertebral junction and cervical spinal cord: pathoanatomical substrate of central cord syndrome and cruciate paralysis. *J. Neurosurg.* 136, 1395–1409 (2022).

REVIEWER COMMENTS

Reviewer #1 (Remarks to the Author):

I feel the authors have addressed my concerns adequately given the constraints in the data collected.

Reviewer #2 (Remarks to the Author):

The authors have addressed many of the previous comments. There are two general issues remaining for further consideration.

1. Proximal to distal

The authors draw parallels between SCI and stroke patients when describing the proximal to distal recovery pattern for upper limb muscles. This pattern makes sense for SCI patients, given the interaction between lesion level and proximal versus distal muscle innervation. However, the evidence for a proximal to distal recovery pattern after stroke is much less clear. The one study cited to support this statement in the Introduction [41] found that proximal muscles had greater recovery than distal muscles only for MEP- patients. There was no proximal-distal difference in MEP+ patients. This is a crucial point not mentioned. Other studies, such as PMID 18571981 Beebe & Lang, found no proximal-distal gradient in the recovery of upper extremity motor deficits in a detailed study of sub-acute stroke patients. The greater likelihood of proximal than distal recovery in MEP- patients relates to the bilateral descending control of proximal musculature, with preservation of ipsilateral descending pathways from the contralesional hemisphere after stroke in MEP- patients. The interaction between lesion location and the somatotopic organisation of descending motor pathways is much less clear in stroke than it is in SCI.

2. Proportional recovery

I remain unconvinced about the application of proportional recovery concepts to this dataset. In the stroke literature, proportional recovery means on average recovering 70% of what was lost on the Upper Extremity Fugl-Meyer scale. Patients who don't have this pattern are said to be "non-fitters" who do not experience "proportional recovery". The authors use the term "proportional recovery" for 22 - 45% for MEP+ patients, zero for AIS A, 37% for AIS B, 24% for AIS C, and 59% for AIS D. This is unhelpful. Put simply, any amount of recovery, from no change to full recovery, can be expressed as a proportion of initial impairment. The authors seem to have simply calculated improvement in score as a percentage of initial impairment, which can range from 0 to 100%, and called all percentages "proportional". At line 634 proportionality is described as "less evident" for AIS B/C, but it's not "less evident" it just a lower proportion. There is some conceptual confusion here that needs to be addressed.

I have concerns about using a proportional recovery approach for individual muscles using a scale with only 0 - 5 points. One of the key criticisms of proportional recovery in the stroke literature is ceiling effects in the Fugl-Meyer scale, with a maximum score of 66. It's hard to imagine how the MMS could provide comparable sensitivity and avoid even greater ceiling effects.

The point of the proportional recovery work in stroke is to identify that some patients recover on average 70% of what was lost ("fitters") and others recover a significantly lower average proportion (non-fitters), and understanding what makes a patient a fitter or non-fitter. The main difference between these two groups is that one group is MEP+ and the other is MEP-, and this underscores the importance of the descending CST. Figure 8b indicates that MEP status does not distinguish between patients who have more or less recovery of initial impairment. In fact, this figure shows there's no clear separation into "fitters" and "non-fitters" that is usually observed in stroke patients. Figure 8d is a bit unhelpful because it obscures any differences in recovery between AIS B/C/D, they are all presented with the same colour. The red dots also potentially obscure blue dots in the bottom right hand corner of this graph. These figures again illustrate that the proportional recovery concept is not well-suited to this dataset.

I disagree with line 561, as there has been no "comparative analysis" as no stroke patients are included in this study, and no direct comparisons made. I also disagree with line 578, as this study did not "discern fitters and non-fitters". The proportions were calculated for the four AIS categories, there doesn't seem to have been an attempt to categorise patients as fitters or non-fitters. Was the proportion of recovery significantly different between the four groups of patients, AIS A/B/C/D?

Line 590 recommends that future studies conduct predictions under the proportional recovery framework. What is the expected accuracy of these predictions?

Line 596 notes that this study could not control the dose and timing of therapy delivered to patients. Therapy dose ought to be measured and included in analyses, not just controlled. This is a potential limitation as it remains unknown how much variance is accounted for by therapy dose.

Perhaps the more original contribution is that baseline total motor score predicts strength recovery in AIS D patients, but MEP status is needed to improve prediction accuracy for AIS A/B/C. This mirrors the stroke literature, in that TMS is not needed to predict upper limb outcomes for patients with mild initial impairment. See PMID 29159193, PMID 28280137. I agree with the authors' final statement, that measures of CST sparing ought to be integrated into clinical assessment following cervical SCI, particularly for AIS A/B/C. I don't think the proportional recovery concept adds anything useful.

Overall, the manuscript has several strengths, however some links to the stroke literature are not well supported and detract from the overall quality of the report, in my view. If the novelty of the report rests on the links to proximal-distal recovery patterns and proportional recovery, then these aspects will need a more sophisticated approach. The general finding that motor recovery of hand muscles is difficult to predict, but predictions improve if MEP status is considered, is a useful observation. This can be linked to the stroke literature without resorting to proportional recovery explanations.

Dear Reviewers:

Thank you for taking the time to review our manuscript and providing feedback that we agree will improve and strengthen our study. We appreciate your feedback, comments, and queries. We have included the Reviewer, Question/Query with our response. It is indicated in the rebuttal letter if the change was incorporated in the revised manuscript. Please note that we decided to keep the R1 changes in red and document all the changes with R2 using track changes.

Note: Please note that to maintain the track changes in the text we had to upload a PDF version of the article. The clean version was also uploaded as a word file and tagged as an article related file.

Reviewer #2 (Remarks to the Author):

The authors have addressed many of the previous comments. There are two general issues remaining for further consideration.

Thank you for taking the time to review once again our manuscript and providing further feedback and reflections that we agree will improve and strengthen our study. Please note that we decided to keep the R1 changes in red and document all the changes with R2 using track changes.

1. Proximal to distal

The authors draw parallels between SCI and stroke patients when describing the proximal to distal recovery pattern for upper limb muscles. This pattern makes sense for SCI patients, given the interaction between lesion level and proximal versus distal muscle innervation. However, the evidence for a proximal to distal recovery pattern after stroke is much less clear. The one study cited to support this statement in the Introduction [41] found that proximal muscles had greater recovery than distal muscles only for MEP- patients. There was no proximal-distal difference in MEP+ patients. This is a crucial point not mentioned. Other studies, such as PMID 18571981 Beebe & Lang, found no proximal-distal gradient in the recovery of upper extremity motor deficits in a detailed study of sub-acute stroke patients. The greater likelihood of proximal than distal recovery in MEP- patients relates to the bilateral descending control of proximal musculature, with preservation of ipsilateral descending pathways from the contralesional hemisphere after stroke in MEP- patients. The interaction between lesion location and the somatotopic organisation of descending motor pathways is much less clear in stroke than it is in SCI.

Answer: We have removed this reference and the portion of the text referring to it. Please see this change on pp. 4, lines 103-105. Because we toned down the presentation of results and discussion about stroke and proportional recovery we refrained to add more text and discussions on this topic. We agree the topic should be explored in future studies with a direct comparison and analysis between datasets obtained from both populations (SCI and stroke). We suggest this on pp. 29, lines 596-598.

2. Proportional recovery

Answer: Overall changes are reflected in the main portions of the manuscript: title, abstract, last paragraph of the introduction, the first paragraph of the discussion and the conclusion of the work. We toned down the presentation of results and discussion about the proportional recovery rule and framed our study as an initial exploration of this phenomenon in SCI. We explored the clustering based on the presence/absence of MEP at baseline and reworded the main conclusion to "Here, we show that post-SCI individuals with MEP+ recover \approx 22-45% of what they have lost – a proportion much lower compared to stroke, without any clear separation into "fitters" and "non-fitters" commonly seen in stroke." This is a preliminary finding we are presenting here, which relates to the importance of using the MEP assessment to quantify the potential of recovery of hand muscles in SCI – a top priority for individuals living with a cervical SCI. We agree with the reviewer that this is the most novel and interesting finding of the study. Finally, we instigate further studies on the proportional recovery rule in SCI using more sophisticated approaches.

Please kindly see the pages and lines of these changes in the point-by-point answer below.

I remain unconvinced about the application of proportional recovery concepts to this dataset. In the stroke literature, proportional recovery means on average recovering 70% of what was lost on the Upper Extremity Fugl-Meyer scale. Patients who don't have this pattern are said to be "non-fitters" who do not experience "proportional recovery". The

authors use the term "proportional recovery" for 22 - 45% for MEP+ patients, zero for AIS A, 37% for AIS B, 24% for AIS C, and 59% for AIS D. This is unhelpful. Put simply, any amount of recovery, from no change to full recovery, can be expressed as a proportion of initial impairment. The authors seem to have simply calculated improvement in score as a percentage of initial impairment, which can range from 0 to 100%, and called all percentages "proportional".

Answer: We have removed the emphasis on the proportional recovery of different AIS severities (AIS A/B/C/D) but maintained the interesting finding about the ability of the MEP at baseline in determining a recovery that is proportional to some extent (regression line slope different from 0). We think that this finding aligns well with the usefulness of the MEP as a predictor for hand strength recovery and constitutes an initial step in understanding the proportional recovery concept in SCI. We suggest that future studies should explore proportional recovery in more detail and use more sophisticated methods – including a direct comparison between stroke and SCI datasets. Please see these changes throughout the manuscript in track changes.

At line 634 proportionality is described as "less evident" for AIS B/C, but it's not "less evident" it just a lower proportion. There is some conceptual confusion here that needs to be addressed.

Answer: We have removed this statement, please see this removal on pp. 32, lines 652-654.

I have concerns about using a proportional recovery approach for individual muscles using a scale with only 0 - 5 points. One of the key criticisms of proportional recovery in the stroke literature is ceiling effects in the Fugl-Meyer scale, with a maximum score of 66. It's hard to imagine how the MMS could provide comparable sensitivity and avoid even greater ceiling effects.

Answer: We have changed this portion of the results to reflect the findings of our non-linear regression using random forest regressors. This analysis supports the proximal to distal gradient in SCI and indicates that motor recovery can be predicted using solely baseline strength for AIS D, with regression lines resembling the proportional recovery seen in stroke. Please see these changes on pp. 9-11, lines 194-195; 197-198; 202-204;

The point of the proportional recovery work in stroke is to identify that some patients recover on average 70% of what was lost ("fitters") and others recover a significantly lower average proportion (non-fitters), and understanding what makes a patient a fitter or non-fitter. The main difference between these two groups is that one group is MEP+ and the other is MEP-, and this underscores the importance of the descending CST. Figure 8b indicates that MEP status does not distinguish between patients who have more or less recovery of initial impairment. In fact, this figure shows there's no clear separation into "fitters" and "non-fitters" that is usually observed in stroke patients. Figure 8d is a bit unhelpful because it obscures any differences in recovery between AIS B/C/D, they are all presented with the same colour. The red dots also potentially obscure blue dots in the bottom right hand corner of this graph. These figures again illustrate that the proportional recovery concept is not well-suited to this dataset.

Answer: We have toned down this portion of the results sections and reframed the findings to reflect the variance seen in the separation between fitters and non-fitters using the MEP, which is much more variable than in stroke. We discuss these in terms of the complexity of the lesion to the spinal cord and the cascade of secondary complications seen in these patients. Please see some additions on pp. 20, lines 388-390. Please note that we report these findings but do not place too much emphasis, we acknowledge the usefulness of the MEP in predicting hand strength recovery, and the ability of the MEP in indicating a proportional recovery (slope different from zero) but without any clear separation into two clusters of individuals (fitters and non-fitters). As previously mentioned, we also suggest that future studies should address this topic using more sophisticated methods and a direct comparison between the recovery profiles of these two populations (SCI and stroke). Please kindly note that overall changes are reflected (track changes) in the main portions of the manuscript: title, abstract, last paragraph of the introduction, the first paragraph of the discussion and the conclusion of the work.

I disagree with line 561, as there has been no "comparative analysis" as no stroke patients are included in this study, and no direct comparisons made.

Answer: We have removed the word "comparative analysis" and added, "discussion concerning". Please see this change on pp. 28, line 570.

I also disagree with line 578, as this study did not "discern fitters and non-fitters". The proportions were calculated for the four AIS categories, there doesn't seem to have been an attempt to categorise patients as fitters or non-fitters. Was the proportion of recovery significantly different between the four groups of patients, AIS A/B/C/D?

Answer: We have reworded this sentence to clarify that although the aim was to obtain two distinct clusters (fitters and non-fitters) by using the MEP as a biomarker - as seen in stroke, the relation was much less evident in SCI. In other words, no clear separation in clusters was observed. Please see this change on pp. 29, lines 587-590; 596-598.

Line 590 recommends that future studies conduct predictions under the proportional recovery framework. What is the expected accuracy of these predictions?

Answer: We have removed the emphasis on proportional recovery and thereby, this portion of the text. All the removals are in track changes on pp. 28 (from line 576) -30 (to line 603).

Line 596 notes that this study could not control the dose and timing of therapy delivered to patients. Therapy dose ought to be measured and included in analyses, not just controlled. This is a potential limitation as it remains unknown how much variance is accounted for by therapy dose.

Answer: We have changed the word "control" to "quantify" on pp. 30, line 609. This limitation is acknowledged in the current version of the manuscript, including the observation on pp. 31, lines 637-638 and the Supplementary Material.

Perhaps the more original contribution is that baseline total motor score predicts strength recovery in AIS D patients, but MEP status is needed to improve prediction accuracy for AIS A/B/C. This mirrors the stroke literature, in that TMS is not needed to predict upper limb outcomes for patients with mild initial impairment. See PMID 29159193, PMID 28280137.

Answer: We have harmonized the text to better reflect these findings. We have added text to reflect the "baseline total motor score predicts strength recovery in AIS D patients" query on pp. 31-32, lines 649-651.

We have added the suggested references on pp.32, lines 658-662.

I agree with the authors' final statement, that measures of CST sparing ought to be integrated into clinical assessment following cervical SCI, particularly for AIS A/B/C.

Answer: We have added "especially for severe paralysis" on pp. 32, line 667. We have toned down the use of emphasis on proportional recovery.

I don't think the proportional recovery concept adds anything useful.

Answer: We have substantially toned down the proportional recovery concept presentation (results) and discussion. Overall changes are reflected in the main portions of the manuscript: title, abstract, last paragraph of the introduction, the first paragraph of the discussion and the conclusion of the work.

Overall, the manuscript has several strengths, however some links to the stroke literature are not well supported and detract from the overall quality of the report, in my view. If the novelty of the report rests on the links to proximal-distal recovery patterns and proportional recovery, then these aspects will need a more sophisticated approach.

Answer: We toned down some links to the stroke literature and propose future studies assessing the links to proximal-distal recovery patterns and proportional recovery. The main strength and novelty of the study lie in the segmental analysis of motor recovery and the importance of the MEP in predicting the recovery of distal hand muscles.

The general finding that motor recovery of hand muscles is difficult to predict, but predictions improve if MEP status is considered, is a useful observation. This can be linked to the stroke literature without resorting to proportional recovery explanations.

Answer: We have strengthened the findings on the motor recovery of hand muscles being difficult to predict with a prediction improvement if MEP status is considered. We have substantially reworked the manuscript in relation to the interpretation and framing of the findings concerning stroke and proportional recovery. The findings on the proportional recovery are placed more conservatively and as a preliminary exploration. We also call for further studies assessing the natural recovery process seen after SCI under the light of the proportional recovery framework developed for stroke with a direct comparison between these neurological populations.

Additional changes:

- Please note we added a recent article on pp. 29, line 594: “Lower Motoneuron Dysfunction Impacts Spontaneous Motor Recovery in Acute Cervical Spinal Cord Injury”.
- Please note we added more in the use of the segmental analysis in upcoming clinical trials, please see this addition on pp. 31, lines 644-646.

REVIEWER COMMENTS

Reviewer #2 (Remarks to the Author):

Thankyou for the thoughtful and thorough response to previous comments. The authors have appropriately de-emphasised prior work on proportional recovery of upper limb impairment after stroke. There are just a couple of minor adjustments that would further clarify the message and improve the flow of ideas.

Line 380: Note that te previous work on proportional recovery in stroke with MEP+ patients is confined to the upper limb. The next two sentences describe the total motor score and then the recovery of the upper limb only for SCI patients. Please make it clear for the reader here, and in the Introduction, that the observation of 70% proportional recovery for MEP+ stroke patients is only for the upper limb as well, not total motor scores. It's also important to point out that the clinical assessment scale used for stroke (UE-FM) is completely different to the scale being used here.

Line 401: MEPs have been used here to predict muscle strength, rather than “spinal cord function”.

Line 437: Please be clear that the recovery is of upper limb strength, not recovery of all general motor function.

Line 450: “Some hand muscles with MEP+ at baseline did not show motor recovery...” Please be more precise. How many hand muscles? Which hand muscles? Does no recovery mean no change at all in strength?

Line 381 reports that SCI patients recover around 22% of what they have lost on the total motor score. Line 385 reports that SCI patients who are MEP+ recover around 45% of what they have lost in the upper limb.

Line 564 of the Discussion notes that “the limited recovery in SCI (22 – 45%) compared to stroke (70%) may reflect the direct lesion to the CST in SCI...”. Are the authors referring to 70% recovery for the upper limb in MEP+ patients? Or 70% recovery of all functions for all stroke patients (ignoring the non-fitters)? The appropriate comparison is upper limb recovery for MEP+ SCI and stroke patients, which seems to be around 45% for the former and 70% for the latter. Please clarify this. Line 568 is unclear – is there scope for compensation in the brain’s sensorimotor system after stroke but not after SCI?

Line 574 – this revised paragraph is much improved. It’s important to also note that the differences between stroke and SCI that produce the lack of fitters and a lower proportional recovery are not just biological, they could also be related to the vastly different clinical assessment scales used. The Upper Extremity Fugl-Meyer scale is entirely different to the strength grading used here, and even with a maximum score of 66 the UE-FM has significant ceiling effects that contribute to the critique of proportional recovery.

There are a couple of uses of the word “proportional” that ought to be removed or re-ordered, for consistency with the revised de-emphasis of this concept. These appear on:

Line 31 – remove proportional

Line 398 – reverse “proportional recovery to” so that it reads “recovery proportional to...”

REVIEWERS' COMMENTS

Reviewer #2 (Remarks to the Author):

Thankyou for the thoughtful and thorough response to previous comments. The authors have appropriately de-emphasised prior work on proportional recovery of upper limb impairment after stroke. There are just a couple of minor adjustments that would further clarify the message and improve the flow of ideas.

Answer: Thanks for the insightful comments and the time dedicated to revising our manuscript, which contributed significantly to improving our work's overall clarity and quality.

Line 380: Note that te previous work on proportional recovery in stroke with MEP+ patients is confined to the upper limb. The next two sentences describe the total motor score and then the recovery of the upper limb only for SCI patients. Please make it clear for the reader here, and in the Introduction, that the observation of 70% proportional recovery for MEP+ stroke patients is only for the upper limb as well, not total motor scores. It's also important to point out that the clinical assessment scale used for stroke (UE-FM) is completely different to the scale being used here.

Answer: Please see these changes in the introduction on pp. 4, line 101 and on pp. 19, line 376. We have also highlighted these differences in the discussion section (on pp. 26, lines 527-529). We have also re-worded and complemented the following sentences in the discussion (on pp. 27-28, lines 570-574):

“It is important to reflect on the differences between stroke and SCI, such as LMN dysfunction, which adds complexity to the lesion – affecting different segments of the spinal cord (depending on the level of SCI); and the clinical assessment scales used for stroke (Fugl-Meyer motor function score: movement, coordination, and reflex) and SCI (ISNCSCI motor score: movement and strength).”

Line 401: MEPs have been used here to predict muscle strength, rather than “spinal cord function”.

Answer: Please see this change on pp. 20, lines 396-397.

Line 437: Please be clear that the recovery is of upper limb strength, not recovery of all general motor function.

Answer: Please see this change on pp. 21, line 433.

Line 450: “Some hand muscles with MEP+ at baseline did not show motor recovery...” Please be more precise. How many hand muscles? Which hand muscles? Does no recovery mean no change at all in strength?

Answer: We have added more information to support this statement, please see this addition on pp. 22, lines 442-443.

Line 381 reports that SCI patients recover around 22% of what they have lost on the total motor score. Line 385 reports that SCI patients who are MEP+ recover around 45% of what they have lost in the upper limb.

Answer: We have added more information to the former sentence (on pp. 19, line 377), which now reads:

“Here, we show that individuals living with SCI with MEP+ recover to a lesser extent, ≈22% of the total motor score with greater variability compared to stroke (*i.e.*, $R^2 = 0.132$).”

Line 564 of the Discussion notes that “the limited recovery in SCI (22 – 45%) compared to stroke (70%) may reflect the direct lesion to the CST in SCI...”. Are the authors referring to 70% recovery for the upper limb in MEP+ patients? Or 70% recovery of all functions for all stroke patients (ignoring the non-fitters)? The appropriate comparison is upper limb recovery for MEP+ SCI and stroke patients, which seems to be around 45% for the former and 70% for the latter. Please clarify this.

Answer: We have clarified this statement on pp. 27, lines 555-556. Now it reads:

“Finally, in individuals with MEP+, the limited upper extremity recovery in SCI (≈45%) compared to stroke (≈70%) may reflect the direct lesion to the CST in SCI, (...).”

Line 568 is unclear – is there scope for compensation in the brain's sensorimotor system after stroke but not after SCI?

Answer: We removed this sentence and references to avoid more extensive additions to the discussion section, which is already very lengthy (currently at 2,896 words).

Line 574 – this revised paragraph is much improved. It's important to also note that the differences between stroke and SCI that produce the lack of fitters and a lower proportional recovery are not just biological, they could also be related to the vastly different clinical assessment scales used. The Upper Extremity Fugl-Meyer scale is entirely different to the strength grading used here, and even with a maximum score of 66 the UE-FM has significant ceiling effects that contribute to the critique of proportional recovery.

Answer: We think that the addition on pp. 28, lines 573-574 highlights these differences to be explored in future studies (pp. 28, lines 575-576).

There are a couple of uses of the word “proportional” that ought to be removed or re-ordered, for consistency with the revised de-emphasis of this concept. These appear on:

Line 31 – remove proportional

Answer: We have changed the word “proportional” for “upper extremity” on pp. 1, lines 29-30. Please also note that the abstract was substantially reduced in size.

Line 398 – reverse “proportional recovery to” so that it reads “recovery proportional to...”

Answer: Please see this change on pp. 20, line 393.